# DECAYING MOMENTUM HELPS NEURAL NETWORK TRAINING

## ABSTRACT

Momentum is a simple and popular technique in deep learning for gradient-based optimizers. We propose a decaying momentum (DEMON) rule, motivated by decaying the total contribution of a gradient to all future updates. Applying DEMON to Adam leads to significantly improved training, notably competitive to momentum SGD with learning rate decay, even in settings in which adaptive methods are typically non-competitive. Similarly, applying DEMON to momentum SGD rivals momentum SGD with learning rate decay, and in some cases leads to improved performance. DEMON is trivial to implement and incurs limited extra computational overhead, compared to the vanilla counterparts.

## 1 INTRODUCTION

Deep Neural Networks (DNNs) have drastically advanced the state-of-the-art performance in many computer science applications, including computer vision (Krizhevsky et al., 2012), (He et al., 2016; Ren et al., 2015), natural language processing (Mikolov et al., 2013; Bahdanau et al., 2014; Gehring et al., 2017) and speech recognition (Sak et al., 2014; Sercu et al., 2016). Yet, in the face of such significant developments, the age-old stochastic gradient descent (SGD), and the accelerated variant SGD with momentum (SGDM), algorithm remains one of the most, if not the most, popular method for training DNNs (Sutskever et al., 2013; Goodfellow et al., 2016; Wilson et al., 2017).

Adaptive methods (Duchi et al., 2011; Zeiler, 2012; Hinton et al., 2012; Kingma & Ba, 2014; Ma & Yarats, 2018) sought to simplify the training process, while providing similar performance. However, while they are often used by practitioners, there are cases where their use leads to a performance gap (Wilson et al., 2017; Shah et al., 2018). At the same time, much of the state-of-the-art performance on highly contested benchmarks—such as the image classification dataset ImageNet—have been produced with SGDM (Krizhevsky et al., 2012; He et al., 2016; Xie et al., 2017; Zagoruyko & Komodakis, 2016; Huang et al., 2017; Ren et al., 2015; Howard et al., 2017).

Nevertheless, a key factor in any algorithmic success still lies in hyperparameter tuning. For example, in the literature above, they obtain such performance with a well-tuned SGD with momentum and a learning rate decay schedule, or with a proper hyperparameter tuning in adaptive methods. Slight changes in learning rate, learning rate decay, momentum, and weight decay (amongst others) can drastically alter performance. Hyperparameter tuning is arguably one of the most time consuming parts of training DNNs, and researchers often resort to a costly grid search. *Thus, finding new and simple hyper-parameter tuning routines that boost the performance of state of the art algorithms is of ultimate importance and one of the most pressing problems in machine learning.*

The focus of this work is on the momentum parameter and how we can boost the performance of training methods with a simple technique. Momentum helps speed up learning in directions of low curvature, without becoming unstable in directions of high curvature. Minimizing the objective function $\mathcal{L}(\cdot)$, the simplest and most common momentum method, SGDM, is given by the following recursion for variable vector $\theta_t \in \mathbb{R}^p$:

$$\theta_{t+1} = \theta_t + \eta v_t, \quad v_t = \beta v_{t-1} - g_t.$$

The coefficient $\beta$—traditionally, selected constant in $[0, 1]$—controls how quickly the momentum decays, $g_t$ represents a stochastic gradient, usually $\mathbb{E}[g_t] = \nabla \mathcal{L}(\theta_t)$, and $\eta > 0$ is the step size.

But how do we select $\beta$? The most prominent choice among practitioners is $\beta = 0.9$. This is supported by recent works that prescribe it (Chen et al., 2016; Kingma & Ba, 2014; Hinton et al.,

2012; Reddi et al., 2019), and by the fact that most common softwares, such as PyTorch (Paszke et al., 2017), declare $\beta = 0.9$ as the default value in their optimizer implementations. However, there is no indication that this choice is universally well-behaved.

There are papers that attempt to tune the momentum parameter. Under an asynchronous distributed setting, (Mitliagkas et al., 2016) observe that running SGD asynchronously is similar to adding a momentum-like term to SGD; they also provide experimental evidence that naively setting $\beta = 0.9$ would result in a momentum "overdose", leading to suboptimal performance. As another example, YellowFin (Zhang & Mitliagkas, 2017) is a learning rate and momentum adaptive method for both the synchronous and asynchronous setting, motivated by a quadratic model analysis and some robustness insights. The main message of that work is that, like $\eta$, momentum acceleration needs to be carefully selected based on properties of the objective, the data, and the underlying computational resources. Finally, moving from classical DNN settings towards generative adversarial networks (GANs), the proposed momentum values tend to decrease from $\beta = 0.9$ (Mirza & Osindero, 2014; Radford et al., 2015; Arjovsky et al., 2017), taking even negative values (Gidel et al., 2018).

In this paper, we introduce a novel momentum decay rule which significantly surpasses the performance of both Adam and SGDM (as they are used currently), in addition to other state-of-the-art adaptive learning rate and adaptive momentum methods, across a variety of datasets and networks. In particular, our findings can be summarized as follows:

$i$) We propose a new momentum decay rule, motivated by decaying the total contribution of a gradient to all future updates, with limited overhead and additional computation.

$ii$) Using the momentum decay rule with Adam, we observe large performance gains—relative to vanilla Adam—where the network continues to learn for far longer after Adam begins to plateau, and suggest that the momentum decay rule should be used as default for this method.

$iii$) We observe comparative performance for SGDM between momentum decay and learning rate decay; an interesting result given the unparalleled effectiveness of learning rate decay schedule.

Experiments are provided on various datasets, including MNIST, CIFAR-10, CIFAR-100, STL-10, Penn Treebank (PTB), and networks, including Convolutional Neural Networks (CNN) with Residual architecture (ResNet) (He et al., 2016), Wide Residual architecture (Wide ResNet) (Zagoruyko & Komodakis, 2016), Non-Residual architecture (VGG-16) (Simonyan & Zisserman, 2014), Recurrent Neural Networks (RNN) with Long Short-Term Memory architecture (LSTM) (Hochreiter & Schmidhuber, 1997), Variational AutoEncoders (VAE) (Kingma & Welling, 2015), and the recent Noise Conditional Score Network (NCSN) (Song & Ermon, 2019).

## 2 PRELIMINARIES

**Plain stochastic gradient descent motions.** Let $\theta_t \in \mathbb{R}^p$ be the parameters of the network at time step $t$, where $\eta \in \mathbb{R}$ is the learning rate/step size, and $g_t$ is the stochastic gradient w.r.t. $\theta_t$ for empirical loss $\mathcal{L}(\cdot)$, such that $\mathbb{E}[g_t] = \nabla \mathcal{L}(\theta_t)$. Then, plain stochastic gradient descent (SGD) uses the recursion: $\theta_{t+1} = \theta_t - \eta \cdot g_t$, $\forall t$. Here, the step size $\eta$ could also be time dependent, $\eta_t$, but practice shows that decreasing the value of $\eta$ at regular or predefined intervals works favorably compared to decreasing the value of $\eta$ at every iteration.

SGDM is parameterized by $\beta \in \mathbb{R}$, the momentum coefficient, and follows the recursion:

$$\theta_{t+1} = \theta_t + \eta v_t, \quad v_t = \beta v_{t-1} - g_t,$$

where $v_t \in \mathbb{R}^p$ accumulates momentum. Observe that for $\beta = 0$, the above recursion is equivalent to SGD. Common values for $\beta$ are closer to one, with $\beta = 0.9$ the most used value (Ruder, 2016).

**Adaptive gradient descent motions.** These algorithms utilize current and past gradient information to design preconditioning matrices that better approximate the local curvature of $\mathcal{L}(\cdot)$. Beginning with AdaGrad (Duchi et al., 2011), the SGD recursion, per coordinate $i$ of $\theta$, becomes:

$$\theta_{t+1,i} = \theta_{t,i} - \frac{\eta}{\sqrt{G_{t,ii} + \varepsilon}} \cdot g_{t,i}, \quad \forall t,$$

where $G_t \in \mathbb{R}^{p \times p}$ is usually a diagonal preconditioning matrix as a summation of squares of past gradients, and $\varepsilon > 0$ a small constant.

RMSprop (Hinton et al., 2012) substitutes the ever accumulating matrix $G_t$ with a root mean squared operation. Denoting the average of squared gradients as $\mathcal{E}_t^{g \circ g}$, per iteration we compute: $\mathcal{E}_{t+1}^{g \circ g} =$

$\beta_2 \cdot \mathcal{E}_t^{g \circ g} + (1 - \beta_2) \cdot (g_t \circ g_t)$, where $\beta_2$ was first proposed as $0.9$. Here, $\circ$ denotes the per-coordinate multiplication. Then, RMSprop updates as—where a momentum term can also be optionally added:

$$\theta_{t+1,i} = \theta_{t,i} - \frac{\eta}{\sqrt{\mathcal{E}_{t+1,i}^{g \circ g} + \varepsilon}} \cdot g_{t,i}, \quad \forall t.$$

Finally, Adam (Kingma & Ba, 2014), in addition, keeps an exponentially decaying average of past gradients: $\mathcal{E}_{t+1}^g = \beta_1 \cdot \mathcal{E}_t^g + (1 - \beta_1) \cdot g_t$, leading to the recursion:[1]

$$\theta_{t+1,i} = \theta_{t,i} - \frac{\eta}{\sqrt{\mathcal{E}_{t+1,i}^{g \circ g} + \varepsilon}} \cdot \mathcal{E}_{t+1,i}^g, \quad \forall t,$$

where usually $\beta_1 = 0.9$ and $\beta_2 = 0.999$. Observe that Adam is equivalent to RMSprop when $\beta_1 = 0$, and when no bias correction is applied results in the same recursion.

## 3 DEMON: DECAYING MOMENTUM ALGORITHM

---

**Algorithm 1** DEMON in SGDM

1: **Parameters**: # of iterations $T$, step size $\eta$, momentum initial value $\beta_{\text{init}}$.
2: $v_t = \theta_t = 0$, otherwise randomly initialized.
3: **for** $t = 0, \dots, T$ **do**
4: $\quad \beta_t = \beta_{\text{init}} \cdot \frac{\left(1 - \frac{t}{T}\right)}{(1 - \beta_{\text{init}}) + \beta_{\text{init}}\left(1 - \frac{t}{T}\right)}$
5: $\quad \theta_{t+1} = \theta_t - \eta g_t + \beta_t v_t$
6: $\quad v_{t+1} = \beta_t v_t - \eta g_t$
7: **end for**

---

**Algorithm 2** DEMON in Adam

1: **Parameters**: # of iterations T, step size $\eta$, momentum initial value $\beta_{\text{init}}$, $\beta_2$, $\varepsilon = 10^{-8}$.
2: $v_t = \theta_t = \mathcal{E}_0^{g \circ g} = 0$, otherwise randomly initialized.
3: **for** $t = 0, \dots, T$ **do**
4: $\quad \beta_t = \beta_{\text{init}} \cdot \frac{\left(1 - \frac{t}{T}\right)}{(1 - \beta_{\text{init}}) + \beta_{\text{init}}\left(1 - \frac{t}{T}\right)}$
5: $\quad \mathcal{E}_{t+1}^{g \circ g} = \beta_2 \cdot \mathcal{E}_t^{g \circ g} + (1 - \beta_2) \cdot (g_t \circ g_t)$
6: $\quad m_{t,i} = g_{t,1} + \beta_t m_{t-1,i}$
7: $\quad \theta_{t+1,i} = \theta_{t,i} - \frac{\eta}{\sqrt{\mathcal{E}_{t+1,i}^{g \circ g} + \varepsilon}} \cdot m_{t,i}$
8: **end for**

---

**Motivation and interpretation.** DEMON is motivated by linear learning rate decay models which reduce the impact of a gradient to current and future updates. By decaying the momentum parameter, we decay the total contribution of a gradient to all future updates. *Our goal here is to present a concrete, effective, and easy-to-use momentum decay procedure which we show in the experimental section.* By Occam's Razor, a linear decay achieves such a goal since it is simple and requires no tuning in most cases. The key component is the momentum decay schedule:

$$\beta_t = \beta_{\text{init}} \cdot \frac{\left(1 - \frac{t}{T}\right)}{(1 - \beta_{\text{init}}) + \beta_{\text{init}}\left(1 - \frac{t}{T}\right)} . \tag{1}$$

Above, the fraction $(1 - t/T)$ refers to the proportion of iterations remaining. The interpretation of this rule comes from the following argument: Assume fixed momentum parameter $\beta_t \equiv \beta$; e.g., $\beta = 0.9$, as literature dictates. For our discussion, we will use the SGDM recursion. We know that $v_0 = 0$, and $v_t = \beta v_{t-1} - g_t$. Then, the main recursion can be unrolled into:

$$\theta_{t+1} = \theta_t - \eta g_t - \eta \beta g_{t-1} - \eta \beta^2 g_{t-2} + \eta \beta^3 v_{t-2} = \cdots = \theta_t - \eta g_t - \eta \cdot \sum_{i=1}^{t} \left(\beta^i \cdot g_{t-i}\right)$$

Interpreting the above recursion, *a particular gradient term $g_t$ contributes a total of $\eta \sum_i \beta^i$ of its "energy" to all future gradient updates.* Moreover, for an asymptotically large number of iterations, we know that $\beta$ contributes on up to $t-1$ terms. Then, $\sum_{i=1}^{\infty} \beta^i = \beta \sum_{i=0}^{\infty} \beta^i = \beta/(1-\beta)$. Thus, in our quest for a decaying schedule and for a simple linear momentum decay, it is natural to consider a scheme where the cumulative momentum is decayed to 0. Let $\beta_{\text{init}}$ be the initial $\beta$; then at current step $t$ with total $T$ steps, we design the decay routine such that: $\beta/(1-\beta) = (1-t/T)\beta_{\text{init}}/(1-\beta_{\text{init}})$. This leads to equation 1. Although $\beta$ changes in subsequent iterations, this is typically a very close approximation since $\beta^i \beta^{i+1} \dots \beta^t$ for a particular $g^i$ diminishes much faster than $\beta$ changes.

**Connection to previous algorithms.** DEMON introduces an implicit discount factor. The main recursions of the algorithm are the same with standard algorithms in machine learning. E.g., for $\beta_t = \beta = 0.9$ we obtain SGD with momentum, and for $\beta = 0$ we obtain plain SGD in Algorithm 1;

---

[1]For clarity, we will skip the bias correction step in this description of Adam; see Kingma & Ba (2014).

in Algorithm 2, for $\beta_1 = 0.9$ with a slightly adjustment of learning rate we obtain Adam, while for $\beta_1 = 0$ we obtain a non-accumulative AdaGrad algorithm. We choose to apply DEMON to a slightly adjusted Adam—instead of vanilla Adam—to isolate the effect of the momentum parameter, since the momentum parameter adjusts the magnitude of the current gradient as well in vanilla Adam.

**Efficiency.** DEMON requires only limited extra overhead and computation in comparison to the vanilla counterparts, for the computation of $\beta_t$.

**Practical suggestions.** For settings in which $\beta_{\text{init}}$ is typically large, such as image classification, we advocate for decaying momentum from $\beta_{\text{init}}$ at $t = 0$, to 0 at $t = T$ as a general rule. We also observe and report improved performance by delaying momentum decay till later epochs. In many cases, performance can be further improved by decaying to a small negative value, such as 0.3.

## 4  RELATED WORK

There are numerous techniques for automatic hyperparameter tuning. The most widely used are learning rate adaptive methods, starting with AdaGrad (Duchi et al., 2011), AdaDelta (Zeiler, 2012), RMSprop (Hinton et al., 2012), and Adam (Kingma & Ba, 2014). Adam (Kingma & Ba, 2014), the most popular, introduced a momentum term, which is combined with the current gradient before multiplying with an adaptive learning rate. Interest in closing the generalization difference between adaptive methods and SGDM led to AMSGrad (Reddi et al., 2019), replacing the exponential moving average of squared gradients with the maximum, QHAdam (Ma & Yarats, 2018), a variant of QHM which is capable of recovering a variety of optimization algorithms, AdamW (Loshchilov & Hutter, 2017), by fixing the weight decay of Adam, and Padam (Chen & Gu, 2018), by lowering the exponent of the second moment.

Asynchronous methods are commonly used in deep learning, and (Mitliagkas et al., 2016) show that running SGD asynchronously is similar to adding a momentum-like term to SGD without assumptions of convexity of the objective function. They demonstrate this natural connection empirically on CNNs. This implies that the momentum parameter needs to be tuned according to the level of asynchrony. YellowFin (Zhang & Mitliagkas, 2017) is a learning rate and momentum adaptive method for both the synchronous and asynchronous setting motivated by a quadratic model analysis and robustness insights. In the non-convex setting, STORM (Cutkosky & Orabona, 2019) uses a variant of momentum for variance reduction.

There is substantial research, both empirical and theoretical, into the convergence of momentum methods (Wibisono & Wilson, 2015; Wibisono et al., 2016; Wilson et al., 2016; Kidambi et al., 2018). In addition, (Sutskever et al., 2013) explored momentum schedules, with even increasing momentum schedules during training, inspired by Nesterov's routines for convex optimization. (Smith et al., 2017) explores increasing batch size instead of decreasing the learning rate, as well as scaling batch size to increasing learning rate or increasing momentum. (Smith, 2018) introduces cycles of simultaneously increasing learning rate and decreasing momentum followed by simultaneously decreasing learning rate and increasing momentum. There is some work into reducing oscillations during training, by adapting the momentum (Odonoghue & Candes, 2015). There is also work into adapting momentum in well-conditioned convex problems as opposed to setting to zero (Srinivasan et al., 2018). Another approach in this area is to keep several momentum vectors according to different $\beta$ and combining them (Lucas et al., 2018). We are aware of the theoretical work of (Yuan, 2016) which prove under certain conditions that SGDM is equivalent to SGD with a rescaled learning rate, however our experiments in the deep learning setting show slightly different behavior and understanding why is an exciting direction of research.

Smaller values of $\beta$ have gradually been employed for Generative Adversarial Networks (GAN), and recent developments in game dynamics (Gidel et al., 2018) show a negative momentum is helpful.

## 5  EXPERIMENTS

We separate experiments into those with adaptive learning rate and those with adaptive momentum. All settings, with exact hyper-parameters, are briefly summarized in Table 1 and comprehensively detailed in Appendix A. We report improved performance by delaying the application of DEMON where applicable, and report performance across different number of total epochs to demonstrate effectiveness regardless of the training budget. Note that the predefined number of epochs we run all experiments affects the proposed decaying momentum routine, by definition of $\beta_t$.

| Experiment short name | Model | Dataset | Optimizer |
|---|---|---|---|
| RN18-CIFAR10-DEMONSGDM | ResNet18 | CIFAR10 | DEMON SGDM |
| RN18-CIFAR10-DEMONAdam | ResNet18 | CIFAR10 | DEMON Adam |
| VGG16-CIFAR100-DEMONSGDM | VGG-16 | CIFAR100 | DEMON SGDM |
| VGG16-CIFAR100-DEMONAdam | VGG-16 | CIFAR100 | DEMON Adam |
| WRN-STL10-DEMONSGDM | Wide ResNet 16-8 | STL10 | DEMON SGDM |
| WRN-STL10-DEMONAdam | Wide ResNet 16-8 | STL10 | DEMON Adam |
| LSTM-PTB-DEMONSGDM | LSTM RNN | Penn TreeBank | DEMON SGDM |
| LSTM-PTB-DEMONAdam | LSTM RNN | Penn TreeBank | DEMON Adam |
| VAE-MNIST-DEMONSGDM | VAE | MNIST | DEMON SGDM |
| VAE-MNIST-DEMONAdam | VAE | MNIST | DEMON Adam |
| NCSN-CIFAR10-DEMONAdam | NCSN | CIFAR10 | DEMON Adam |

Table 1: Summary of experimental settings.

| | 30 epochs | 75 epochs | 150 epochs | 300 epochs |
|---|---|---|---|---|
| Adam | $16.58 \pm .18$ | $13.63 \pm .22$ | $11.90 \pm .06$ | $11.94 \pm .06$ |
| AMSGrad | $16.98 \pm .36$ | $13.43 \pm .14$ | $11.83 \pm .12$ | $10.48 \pm .12$ |
| QHAdam | $16.41 \pm .38$ | $15.55 \pm .25$ | $13.78 \pm .08$ | $13.36 \pm .11$ |
| DEMON Adam | $\mathbf{11.75} \pm .15$ | $\mathbf{9.69} \pm .10$ | $\mathbf{8.83} \pm .08$ | $\mathbf{8.44} \pm .05$ |

Table 2: `RN18-CIFAR10-DEMONAdam` generalization error. The number of epochs was predefined before the execution of the algorithms.

## 5.1 ADAPTIVE METHODS

At first, we apply DEMON Adam (Algorithm 2) to a variety of models and tasks. We select vanilla Adam as the baseline algorithm and include more recent state-of-the-art adaptive learning rate methods Quasi-Hyperbolic Adam (QHAdam) (Ma & Yarats, 2018) and AMSGrad (Reddi et al., 2019) in our comparison. See Appendix A.2.1 for details. We tune all learning rates in roughly multiples of 3 and try to keep all other parameters close to those recommended in the original literature. For DEMON Adam, we leave $\beta_{\text{init}} = 0.9, \beta_2 = 0.999$ and decay from $\beta_{\text{init}}$ to 0 in all experiments.

**Residual Neural Network** (`RN18-CIFAR10-DEMONAdam`). We train a ResNet18 (He et al., 2016) model on the CIFAR-10 dataset. With DEMON Adam, we achieve the generalization error reported in the literature (He et al., 2016) for this model, attained using SGDM and a curated learning rate decay schedule, whilst all other methods are non-competitive. Refer to Table 2 for exact results.

In Figure 2 (*Top row, two left-most plots*), DEMON Adam is able to learn in terms of both loss and accuracy after other methods have plateaued. Running 5 seeds, DEMON Adam outperforms all other methods by a large 2%-5% generalization error margin with a small and large number of epochs.

**Non-Residual Neural Network** (`VGG16-CIFAR100-DEMONAdam`). For the CIFAR-100 dataset, we train an adjusted VGG-16 model (Simonyan & Zisserman, 2014). Similarly to the previous setting, we observe similar learning behavior of DEMON Adam, where it continues to improve after other methods appear to begin to plateau. We note that this behavior results in a 1-3% decrease in generalization error than typically reported results with the same model and task (Sankaranarayanan et al., 2018), which are attained using SGDM and a curated learning rate decay schedule.

Running 5 seeds, DEMON Adam achieves an improvement of 3%-6% generalization error margin over all other methods, both for a small and large number of epochs. Refer to Figure 2 (*Top row, right-most plot*) and Table 3 for more details.

**Wide Residual Neural Network** (`WRN-STL10-DEMONAdam`). The STL-10 dataset presents a different challenge with a significantly smaller number of images than the CIFAR datasets, but in higher resolution. We train a Wide Residual 16-8 model (Zagoruyko & Komodakis, 2016) for this task. In this setting, we note again the behavior of DEMON Adam significantly outperforming other methods in the latter stages of training.

Running 5 seeds, DEMON Adam outperforms all other methods by a 0.5%-2% generalization error margin with a small and large number of epochs. Refer to Figure 2 (*Bottom row, left-most plot*) and Table 3 for more details.

|  | **VGG-16** | | | **Wide Residual 16-8** | | |
|---|---|---|---|---|---|---|
|  | 75 epochs | 150 epochs | 300 epochs | 50 epochs | 100 epochs | 200 epochs |
| Adam | $37.98 \pm .20$ | $33.62 \pm .11$ | $31.09 \pm .09$ | $23.35 \pm .20$ | $19.63 \pm .26$ | $18.65 \pm .07$ |
| AMSGrad | $40.67 \pm .65$ | $34.46 \pm .21$ | $31.62 \pm .12$ | $21.73 \pm .25$ | $19.35 \pm .20$ | $18.21 \pm .18$ |
| QHAdam | $36.53 \pm .20$ | $32.96 \pm .11$ | $30.97 \pm .10$ | $21.25 \pm .22$ | $19.81 \pm .18$ | $18.52 \pm .25$ |
| DEMON Adam | $\mathbf{32.40} \pm .19$ | $\mathbf{28.84} \pm .18$ | $\mathbf{27.11} \pm .19$ | $\mathbf{19.42} \pm .10$ | $\mathbf{18.36} \pm .11$ | $\mathbf{17.62} \pm .12$ |

Table 3: `VGG16-CIFAR100-DEMONAdam` and `WRN-STL10-DEMONAdam` generalization error. The number of epochs was predefined before the execution of the algorithms.

|  | **LSTM** | | **VAE** | | | **NCSN** |
|---|---|---|---|---|---|---|
|  | 25 epochs | 39 epochs | 50 epochs | 100 epochs | 200 epochs | 512 epochs |
| Adam | $115.54 \pm .64$ | $115.02 \pm .52$ | $136.28 \pm .18$ | $134.64 \pm .14$ | $134.66 \pm .17$ | $\mathbf{8.15} \pm .20$ |
| AMSGrad | $108.07 \pm .19$ | $107.87 \pm .25$ | $137.89 \pm .12$ | $135.69 \pm .03$ | $134.75 \pm .18$ | - |
| QHAdam | $112.52 \pm .23$ | $112.45 \pm .39$ | $136.69 \pm .17$ | $134.84 \pm .08$ | $134.12 \pm .12$ | - |
| DEMON Adam | $\mathbf{101.57} \pm .32$ | $\mathbf{101.44} \pm .47$ | $\mathbf{134.46} \pm .17$ | $\mathbf{134.12} \pm .08$ | $\mathbf{133.87} \pm .21$ | $8.07 \pm .08$ |

Table 4: `PTB-LSTM-DEMONAdam` generalization perplexity, `VAE-MNIST-DEMONAdam` generalization loss and `NCSN-CIFAR10-DEMONAdam` inception score.

**LSTM** (`PTB-LSTM-DEMONAdam`). Language modeling can have gradient distributions which are sharp; for example, in the case of rare words. We use an LSTM (Hochreiter & Schmidhuber, 1997) model to this task. We observe overfitting for all adaptive methods.

Similar to above, running 5 seeds, DEMON Adam outperforms all other methods by a *6-14 generalization perplexity margin*. Refer to Figure 2 (*Bottom row, middle plot*) and Table 4 for details.

**Variational AutoEncoder** (`VAE-MNIST-DEMONAdam`). Generative models are a branch of unsupervised learning that try to learn the data distribution. VAEs (Kingma & Welling, 2015) pair a generator network with a second Neural Network, a recognition model that performs approximate inference, and can be trained with backpropagation. We train VAEs on the MNIST dataset.

Running 5 seeds, DEMON Adam outperforms all other methods, particularly for smaller number of epochs. Refer to Figure 2 (*Bottom row, right-most plot*) and Table 4 for more details.

**Noise Conditional Score Network** (`NCSN-CIFAR10-DEMONAdam`). NCSN (Song & Ermon, 2019) is a recent generative network achieving state-of-the-art inception score on CIFAR10. NCSN estimates the gradients of the data distribution with score matching. Samples are then produced via Langevin dynamics using those gradients. We train a NCSN on the CIFAR10 dataset and, using the official implementation, were unable to reproduce the reported score in the literature. NSCN trained with Adam achieves a superior inception score in Table 4, however the produced images in Figure 1 exhibit a noticeably unnatural green compared to those produced by DEMON Adam.

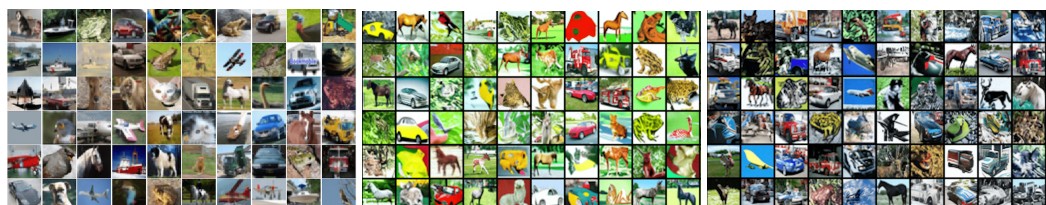

Figure 1: Randomly selected CIFAR10 images generated with NCSN. Left: Real CIFAR10 images. Middle: Adam. Right: DEMON Adam.

## 5.2 ADAPTIVE MOMENTUM METHODS

We apply DEMON SGDM (Algorithm 1) to a variety of models and tasks. Since SGDM with learning rate decay is most often used to achieve the state-of-the-art results with the architectures and tasks in question, we include SGDM with learning rate decay as the target to beat. SGDM with learning rate decay is implemented with a decay on validation error plateau, where we hand-tune the number of epochs to define plateau. Recent adaptive momentum methods included in this section are Aggregated Momentum (AggMo) (Lucas et al., 2018), and Quasi-Hyperbolic Momentum

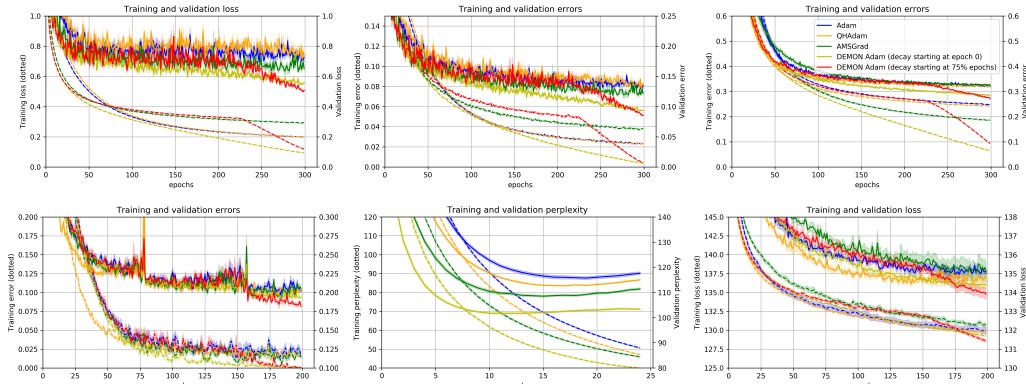

Figure 2: Top row, two left-most plots: `RN18-CIFAR10-DEMONAdam` for 300 epochs. Top row, right-most plot: `VGG16-CIFAR100-DEMONAdam` for 300 epochs. Bottom row, left-most plot: `WRN-STL10-DEMONAdam` for 200 epochs. Bottom row, middle plot: `PTB-LSTM-DEMONAdam` for 25 epochs. Bottom row, right-most plot: `VAE-MNIST-DEMONAdam` for 200 epochs. Dotted and solid lines represent training and generalization metrics respectively. Shaded bands represent one standard deviation.

|  | 30 epochs | 75 epochs | 150 epochs | 300 epochs |
|---|---|---|---|---|
| SGDM learning rate decay | $11.29 \pm .35$ | $9.05 \pm .07$ | $\mathbf{8.26} \pm .07$ | $7.97 \pm .14$ |
| AggMo | $18.85 \pm .27$ | $13.02 \pm .23$ | $11.95 \pm .15$ | $10.94 \pm .12$ |
| QHM | $14.65 \pm .24$ | $12.66 \pm .19$ | $11.27 \pm .13$ | $10.42 \pm .05$ |
| DEMON SGDM | $\mathbf{10.89} \pm .12$ | $\mathbf{8.97} \pm .16$ | $8.39 \pm .10$ | $\mathbf{7.58} \pm .04$ |

Table 5: `RN18-CIFAR10-DEMONSGDM` generalization error. The number of epochs was predefined before the execution of the algorithms.

(QHM) (Ma & Yarats, 2018). We exclude accelerated SGD (Jain et al., 2017) due to difficulties in tuning. See Appendix A.2.2 for details. Similar to the last section, we tune all learning rates in roughly multiples of 3 and try to keep all other parameters close to those recommended in the original literature. For DEMON SGDM, we leave $\beta_{\text{init}} = 0.9$ for most experiments and generally decay from $\beta_{\text{init}}$ to 0.

**Residual Neural Network** (`RN18-CIFAR10-DEMONSGDM`). We train a ResNet18 model on the CIFAR-10 dataset. With DEMON SGDM, we achieve better generalization error than SGDM with learning rate decay, the optimizer for producing state-of-the-art results with ResNet architecture. The better performance of decaying momentum relative to learning rate decay is surprising.

Running 5 seeds, DEMON SGDM outperforms all other adaptive momentum methods by a large 3%-8% validation error margin with a small and large number of epochs and is competitive or better than SGDM with learning rate decay. In Figure 3 (*Top row, two left-most plots*), DEMON SGDM is observed to continue learning after other adaptive momentum methods appear to begin to plateau.

**Non-Residual Neural Network** (`VGG16-CIFAR100-DEMONSGDM`). For the CIFAR-100 dataset, we train an adjusted VGG-16 model. In Figure 3 (*Top row, right-most plot*), we observe DEMON SGDM to learn slowly initially in loss and error, but similar to the previous setting it continues to learn after other methods begin to plateau, resulting in superior final generalization error.

Running 5 seeds, DEMON SGDM achieves an improvement of 1%-8% generalization error margin over all other methods. Refer to Table 6 for more details.

**Wide Residual Neural Network** (`WRN-STL10-DEMONSGDM`). We train a Wide Residual 16-8 model for the STL-10 dataset. In Figure 3 (*Bottom row, left-most plot*), training in both loss and error slows down quickly for other adaptive momentum methods with a large gap with SGDM learning rate decay. DEMON SGDM continues to improve and eventually catches up to SGDM learning rate decay.

Running 5 seeds, DEMON SGDM outperforms all other methods by a 1.5%-2% generalization error margin with a small and large number of epochs. Refer to Table 6 for more details.

| | VGG-16 | | | Wide Residual 16-8 | | |
|---|---|---|---|---|---|---|
| | 75 epochs | 150 epochs | 300 epochs | 75 epochs | 150 epochs | 300 epochs |
| SGDM learning rate decay | 35.29 ± .59 | 30.65 ± .31 | 29.74 ± .43 | 21.05 ± .27 | 17.83 ± 0.39 | 15.16 ± .36 |
| AggMo | 42.85 ± .89 | 34.25 ± .24 | 32.32 ± .18 | 22.70 ± .11 | 20.06 ± .31 | 17.90 ± .13 |
| QHM | 42.14 ± .79 | 33.87 ± .26 | 32.45 ± .13 | 22.86 ± .15 | 19.40 ± .23 | 17.79 ± .08 |
| DEMON SGDM | **34.35** ± .44 | **30.59** ± .26 | **28.99** ± .16 | **19.45** ± .20 | **15.98** ± .40 | **13.67** ± .13 |

Table 6: `VGG16-CIFAR100-DEMONSGDM` and `WRN-STL10-DEMONSGDM` generalization error. The number of epochs was predefined before the execution.

| | LSTM | | VAE | | |
|---|---|---|---|---|---|
| | 25 epochs | 39 epochs | 50 epochs | 100 epochs | 200 epochs |
| SGDM learning rate decay | 89.59 ± .07 | **87.57** ± .11 | 140.51 ± .73 | 139.54 ± .34 | 137.33 ± .49 |
| AggMo | 89.09 ± .16 | 89.07 ± .15 | 139.69 ± .17 | 139.07 ± .26 | 137.64 ± .20 |
| QHM | 94.47 ± .19 | 94.44 ± .13 | 145.84 ± .39 | 140.92 ± .19 | 137.64 ± .20 |
| DEMON SGDM | **88.33** ± .16 | 88.32 ± .12 | **139.32** ± .23 | **137.51** ± .29 | **135.95** ± .21 |

Table 7: `PTB-LSTM-DEMONSGDM` (perplexity) and `VAE-MNIST-DEMONSGDM` (generalization loss) experiments.

**LSTM** (`PTB-LSTM-DEMONSGDM`). We train an RNN with LSTM architecture for the PTB language modeling task. Running 5 seeds, DEMON SGDM slightly outperforms other adaptive momentum methods in generalization perplexity, and is competitive with SGDM with learning rate decay. Refer to Figure 3 (*Bottom row, middle plot*) and Table 7 for more details.

**Variational AutoEncoder** (`VAE-MNIST-DEMONSGDM`). We train the generative model VAE on the MNIST dataset. Running 5 seeds, DEMON SGDM outperforms all other methods by a 2%-6% generalization error for a small and large number of epochs. Refer to Figure 3 (*Bottom row, right-most plot*) and Table 7 for more details.

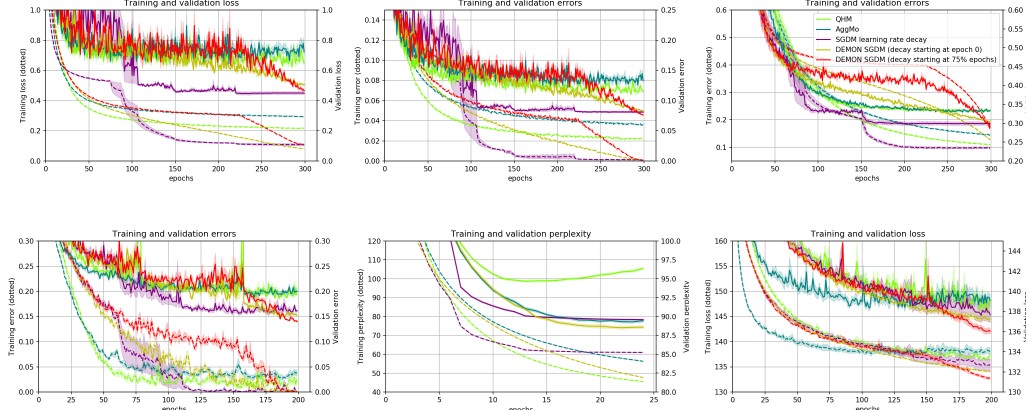

Figure 3: Top row, two left-most plot: `RN18-CIFAR10-DEMONSGDM` for 300 epochs. Top row, right-most plot: `VGG16-CIFAR100-DEMONSGDM` for 300 epochs. Bottom row, left-most plots: `WRN-STL10-DEMONSGDM` for 200 epochs. Bottom row, middle plot: `PTB-LSTM-DEMONSGDM` for 25 epochs. Bottom row, right-most plot: `VAE-MNIST-DEMONSGDM` for 200 epochs. Dotted and solid lines represent training and generalization metrics respectively. Shaded bands represent 1 standard deviation.

## 6 CONCLUSION

We show the effectiveness of the proposed momentum decay rule, DEMON, across a number of datasets and architectures. The adaptive optimizer Adam combined with DEMON is empirically substantially superior to the popular Adam, in addition to other state-of-the-art adaptive learning rate algorithms, suggesting a drop-in replacement. Surprisingly, it is also demonstrated that DEMON SGDM is comparable to SGDM with learning rate decay. In cases where budget is limited, DEMON SGDM may be preferable. DEMON is computationally cheap, easy to understand and use, and we hope it is useful in practice and as a subject of future research.

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

## A  EXPERIMENTS

We evaluated the momentum decay rule with Adam and SGDM on Residual CNNs, Non Residual CNNS, RNNs and generative models. For CNNs, we used the image classification datasets CI-FAR10, CIFAR100 and STL10 datasets. For RNNs, we used the language modeling dataset PTB. For generative modeling, we used the MNIST and CIFAR10 datasets. For each network dataset pair other than NSCN, we evaluated Adam, QHAdam, AMSGrad, DEMON Adam, AggMo, QHM, DE-MON SGDM, and SGDM with learning rate decay. For adaptive learning rate methods and adaptive momentum methods, we generally perform a grid search over the learning rate. For SGDM, we generally perform a grid search over learning rate and initial momentum. For SGDM learning rate decay, the learning rate is decayed by a factor of 0.1 after there is no improvement in validation loss for the best of $\{1, 2, 3, 5, 10, 20, 30, 40\}$ epochs.

### A.1  SETUP

We describe the six test problems in this paper.

- **CIFAR10 - ResNet18** CIFAR10 contains 60,000 32x32x3 images with a 50,000 training set, 10,000 test set split. There are 10 classes. ResNet18 (He et al., 2016) is an 18 layers deep CNN with skip connections for image classification. Trained with a batch size of 128.

- **CIFAR100 - VGG16** CIFAR100 is a fine-grained version of CIFAR-10 and contains 60,000 32x32x3 images with a 50,000 training set, 10,000 test set split. There are 100 classes. VGG16 (Simonyan & Zisserman, 2014) is a 16 layers deep CNN with extensive use of 3x3 convolutional filters. Trained with a batch size of 128

- **STL10 - Wide ResNet 16-8** STL10 contains 1300 96x96x3 images with a 500 training set, 800 test set split. There are 10 classes. Wide ResNet 16-8 (Zagoruyko & Komodakis, 2016) is a 16 layers deep ResNet which is 8 times wider. Trained with a batch size of 64.

- **PTB - LSTM** PTB is an English text corpus containing 929,000 training words, 73,000 validation words, and 82,000 test words. There are 10,000 words in the vocabulary. The model is stacked LSTMs (Hochreiter & Schmidhuber, 1997) with 2 layers, 650 units per layer, and dropout of 0.5. Trained with a batch size of 20.

- **MNIST - VAE** MNIST contains 60,000 32x32x1 grayscale images with a 50,000 training set, 10,000 test set split. There are 10 classes of 10 digits. VAE (Kingma & Welling, 2015) with three dense encoding layers and three dense decoding layers with a latent space of size 2. Trained with a batch size of 100.

- **CIFAR10 - NCSN** CIFAR10 contains 60,000 32x32x3 images with a 50,000 training set, 10,000 test set split. There are 10 classes. NCSN (Song & Ermon, 2019) is a recent state-of-the-art generative model which achieves the best reported inception score. We compute inception scores based on a total of 50000 samples. We follow the exact implementation in and defer details to the original paper.

### A.2  METHODS

#### A.2.1  ADAPTIVE LEARNING RATE

**Adam** (Kingma & Ba, 2014), as previously introduced in section 2, keeps an exponentially decaying average of squares of past gradients to adapt the learning rate. It also introduces an exponentially decaying average of gradients.

The Adam algorithm is parameterized by learning rate $\eta > 0$, discount factors $\beta_1 < 1$ and $\beta_2 < 1$, a small constant $\epsilon$, and uses the update rule:

$$
\begin{aligned}
\mathcal{E}_{t+1}^g &= \beta_1 \cdot \mathcal{E}_t^g + (1 - \beta_1) \cdot g_t, \\
\mathcal{E}_{t+1}^{g \circ g} &= \beta_2 \cdot \mathcal{E}_t^{g \circ g} + (1 - \beta_2) \cdot (g_t \circ g_t), \\
\theta_{t+1,i} &= \theta_{t,i} - \frac{\eta}{\sqrt{\mathcal{E}_{t+1,i}^{g \circ g} + \varepsilon}} \cdot \mathcal{E}_{t+1,i}^g, \quad \forall t.
\end{aligned}
$$

**AMSGrad** (Reddi et al., 2019) resolves an issue in the proof of Adam related to the exponential moving average $\mathcal{E}_t^{g \circ g}$, where Adam does not converge for a simple optimization problem. Instead of an exponential moving average, AMSGrad keeps a running maximum of $\mathcal{E}^{g \circ g}$.

The AMSGrad algorithm is parameterized by learning rate $\eta > 0$, discount factors $\beta_1 < 1$ and $\beta_2 < 1$, a small constant $\epsilon$, and uses the update rule:

$$\begin{aligned}
\mathcal{E}_{t+1}^g &= \beta_1 \cdot \mathcal{E}_t^g + (1 - \beta_1) \cdot g_t, \\
\mathcal{E}_{t+1}^{g \circ g} &= \beta_2 \cdot \mathcal{E}_t^{g \circ g} + (1 - \beta_2) \cdot (g_t \circ g_t), \\
\hat{\mathcal{E}}_{t+1,i}^{g \circ g} &= max(\hat{\mathcal{E}}_{t,i}^{g \circ g}, \mathcal{E}_{t,i}^{g \circ g}), \\
\theta_{t+1,i} &= \theta_{t,i} - \frac{\eta}{\sqrt{\hat{\mathcal{E}}_{t+1,i}^{g \circ g} + \varepsilon}} \cdot \mathcal{E}_{t+1,i}^g, \quad \forall t,
\end{aligned}$$

where $\mathcal{E}_{t+1}^g$ and $\mathcal{E}_{t+1}^{g \circ g}$ are defined identically to Adam.

**QHAdam** (Quasi-Hyperbolic Adam) (Ma & Yarats, 2018) extends QHM (Quasi-Hyperbolic Momentum), introduced further below, to replace both momentum estimators in Adam with quasi-hyperbolic terms. This quasi-hyperbolic formulation is capable of recovering Adam and NAdam (Dozat, 2016), amongst others.

The QHAdam algorithm is parameterized by learning rate $\eta > 0$, discount factors $\beta_1 < 1$ and $\beta_2 < 1$, $\nu_1, \nu_2 \in \mathbb{R}$, a small constant $\epsilon$, and uses the update rule:

$$\begin{aligned}
\mathcal{E}_{t+1}^g &= \beta_1 \cdot \mathcal{E}_t^g + (1 - \beta_1) \cdot g_t, \\
\mathcal{E}_{t+1}^{g \circ g} &= \beta_2 \cdot \mathcal{E}_t^{g \circ g} + (1 - \beta_2) \cdot (g_t \circ g_t), \\
\hat{\mathcal{E}}_{t+1}^g &= (1 + \beta_1^{t+1})^{-1} \cdot \mathcal{E}_{t+1}^g, \\
\hat{\mathcal{E}}_{t+1}^{g \circ g} &= (1 + \beta_2^{t+1})^{-1} \cdot \mathcal{E}_{t+1}^{g \circ g}, \\
\theta_{t+1,i} &= \theta_{t,i} - \eta \left[ \frac{(1 - \nu_1) \cdot g_t + \nu_1 \cdot \hat{\mathcal{E}}_{t+1}^g}{\sqrt{(1 - \nu_2) g_t^2 + \nu_2 \cdot \hat{\mathcal{E}}_{t+1}^{g \circ g} + \epsilon}} \right], \quad \forall t,
\end{aligned}$$

where $\mathcal{E}_{t+1}^g$ and $\mathcal{E}_{t+1}^{g \circ g}$ are defined identically to Adam.

### A.2.2 Adaptive momentum

**AggMo** (Aggregated Momentum) (Lucas et al., 2018) takes a linear combination of multiple momentum buffers. It maintains $K$ momentum buffers, each with a different discount factor, and averages them for the update.

The AggMo algorithm is parameterized by learning rate $\eta > 0$, discount factors $\beta \in \mathbb{R}^K$, and uses the update rule:

$$\begin{aligned}
(\mathcal{E}_{t+1}^g)^{(i)} &= \beta^{(i)} \cdot (\mathcal{E}_t^g)^{(i)} + g_t, \quad \forall i \in [1, K], \\
\theta_{t+1,i} &= \theta_{t,i} - \eta \left[ \frac{1}{K} \cdot \sum_{i=1}^{K} (\mathcal{E}_{t+1}^g)^{(i)} \right], \quad \forall t.
\end{aligned}$$

**QHM** (Quasi-Hyperbolic Momentum) (Ma & Yarats, 2018) is a weighted average of the momentum and plain SGD. QHM is capable of recovering Nesterov Momentum (Nesterov, 1983), Synthesized Nesterov Variants (Lessard et al., 2016), accSGD (Jain et al., 2017) and others.

The QHM algorithm is parameterized by learning rate $\eta > 0$, discount factor $\beta < 1$, immediate discount factor $\nu \in \mathbb{R}$, and uses the update rule:

$$\begin{aligned}
\mathcal{E}_{t+1}^g &= \beta \cdot \mathcal{E}_t^g + (1 - \beta) \cdot g_t, \\
\theta_{t+1,i} &= \theta_{t,i} - \eta \left[ (1 - \nu) \cdot g_t + \nu \cdot \mathcal{E}_{t+1}^g \right], \quad \forall t.
\end{aligned}$$

## A.3 OPTIMIZER HYPERPARAMETERS

Table 8: Best parameters for CIFAR-10 with ResNet-18.

| Optimization method | epochs | $\eta$ | other parameters |
|---|---|---|---|
| Adam | 30 | 0.001 | |
| Adam | 75 | 0.001 | $\beta_1 = 0.9, \beta_2 = 0.999$ |
| Adam | 150 | 0.001 | |
| Adam | 300 | 0.0003 | |
| AMSGrad | 30 | 0.001 | |
| AMSGrad | 75 | 0.001 | $\beta_1 = 0.9, \beta_2 = 0.999$ |
| AMSGrad | 150 | 0.001 | |
| AMSGrad | 300 | 0.001 | |
| QHAdam | 30 | 0.001 | |
| QHAdam | 75 | 0.0003 | $\nu_1 = 0.7, \nu_2 = 1.0, \beta_1 = 0.9, \beta_2 = 0.99$ |
| QHAdam | 150 | 0.0003 | |
| QHAdam | 300 | 0.0003 | |
| DEMON Adam | 30 | 0.0001 | |
| DEMON Adam | 75 | 0.0001 | $\beta_{\text{init}} = 0.9, \beta_2 = 0.999$ |
| DEMON Adam | 150 | 0.0001 | |
| DEMON Adam | 300 | 0.0001 | |
| AggMo | 30 | 0.03 | |
| AggMo | 75 | 0.01 | $\beta = [0, 0.9, 0.99]$ |
| AggMo | 150 | 0.01 | |
| AggMo | 300 | 0.01 | |
| QHM | 30 | 1.0 | |
| QHM | 75 | 0.3 | $\nu = 0.7, \beta = 0.999$ |
| QHM | 150 | 0.3 | |
| QHM | 300 | 0.3 | |
| DEMON SGDM | 30 | 0.1 | |
| DEMON SGDM | 75 | 0.1 | $\beta_{\text{init}} = 0.9$ |
| DEMON SGDM | 150 | 0.03 | |
| DEMON SGDM | 300 | 0.03 | |
| SGDM learning rate decay | 30 | 0.1 | $\beta_1 = 0.9$, patience = 5 |
| SGDM learning rate decay | 75 | 0.1 | $\beta_1 = 0.9$, patience = 20 |
| SGDM learning rate decay | 150 | 0.1 | $\beta_1 = 0.9$, patience = 20 |
| SGDM learning rate decay | 300 | 0.1 | $\beta_1 = 0.9$, patience = 40 |

Table 9: Best parameters for CIFAR-100 with VGG-16.

| Optimization method | epochs | $\eta$ | other parameters |
|---|---|---|---|
| Adam | 75 | 0.0003 | |
| Adam | 150 | 0.0003 | $\beta_1 = 0.9, \beta_2 = 0.999$ |
| Adam | 300 | 0.0003 | |
| AMSGrad | 75 | 0.0003 | |
| AMSGrad | 150 | 0.0003 | $\beta_1 = 0.9, \beta_2 = 0.999$ |
| AMSGrad | 300 | 0.0003 | |
| QHAdam | 75 | 0.0003 | |
| QHAdam | 150 | 0.0003 | $\nu_1 = 0.7, \nu_2 = 1.0, \beta_1 = 0.9, \beta_2 = 0.99$ |
| QHAdam | 300 | 0.0003 | |
| DEMON Adam | 75 | 0.00003 | |
| DEMON Adam | 150 | 0.00003 | $\beta_{\text{init}} = 0.9, \beta_2 = 0.999$ |
| DEMON Adam | 300 | 0.00003 | |
| AggMo | 75 | 0.001 | |
| AggMo | 150 | 0.001 | $\beta = [0, 0.9, 0.99]$ |
| AggMo | 300 | 0.001 | |
| QHM | 75 | 0.1 | |
| QHM | 150 | 0.03 | $\nu = 0.7, \beta = 0.999$ |
| QHM | 300 | 0.03 | |
| DEMON SGDM | 75 | 0.1 | |
| DEMON SGDM | 150 | 0.03 | $\beta_{\text{init}} = 0.9$ |
| DEMON SGDM | 300 | 0.03 | |
| SGDM learning rate decay | 75 | 0.1 | $\beta_1 = 0.9$, patience = 5 |
| SGDM learning rate decay | 150 | 0.03 | $\beta_1 = 0.9$, patience = 20 |
| SGDM learning rate decay | 300 | 0.03 | $\beta_1 = 0.9$, patience = 30 |

Table 10: Best parameters for STL10 with Wide ResNet 16-8.

| Optimization method | epochs | $\eta$ | |
|---|---|---|---|
| Adam | 50 | 0.001 | |
| Adam | 100 | 0.0003 | $\beta_1 = 0.9, \beta_2 = 0.999$ |
| Adam | 200 | 0.0003 | |
| AMSGrad | 50 | 0.0003 | |
| AMSGrad | 100 | 0.0003 | $\beta_1 = 0.9, \beta_2 = 0.999$ |
| AMSGrad | 200 | 0.0003 | |
| QHAdam | 50 | 0.0003 | |
| QHAdam | 100 | 0.0003 | $\nu_1 = 0.7, \nu_2 = 1.0, \beta_1 = 0.9, \beta_2 = 0.99$ |
| QHAdam | 200 | 0.0003 | |
| DEMON Adam | 50 | 0.00003 | |
| DEMON Adam | 100 | 0.00003 | $\beta_{\text{init}} = 0.9, \beta_2 = 0.999$ |
| DEMON Adam | 200 | 0.00003 | |
| AggMo | 50 | 0.03 | |
| AggMo | 100 | 0.03 | $\beta = [0, 0.9, 0.99]$ |
| AggMo | 200 | 0.01 | |
| QHM | 50 | 0.3 | |
| QHM | 100 | 0.3 | $\nu = 0.7, \beta = 0.999$ |
| QHM | 200 | 0.3 | |
| DEMON SGDM | 50 | 0.1 | |
| DEMON SGDM | 100 | 0.1 | $\beta_{\text{init}} = 0.9$ |
| DEMON SGDM | 200 | 0.1 | |
| SGDM learning rate decay | 50 | 0.1 | $\beta_1 = 0.9$, patience = 10 |
| SGDM learning rate decay | 100 | 0.1 | $\beta_1 = 0.9$, patience = 10 |
| SGDM learning rate decay | 200 | 0.1 | $\beta_1 = 0.9$, patience = 20 |

Table 11: Best parameters for PTB with LSTM architecture.

| Optimization method | epochs | $\eta$ | other parameters |
|---|---|---|---|
| Adam | 25 | 0.0003 | $\beta_1 = 0.9, \beta_2 = 0.999$ |
| Adam | 39 | 0.0003 | |
| AMSGrad | 25 | 0.001 | $\beta_1 = 0.9, \beta_2 = 0.999$ |
| AMSGrad | 39 | 0.001 | |
| QHAdam | 25 | 0.0003 | $\nu_1 = 0.7, \nu_2 = 1.0, \beta_1 = 0.9, \beta_2 = 0.999$ |
| QHAdam | 39 | 0.0003 | |
| DEMON Adam | 25 | 0.0001 | $\beta_{\text{init}} = 0.9, \beta_2 = 0.999$ |
| DEMON Adam | 39 | 0.0001 | |
| AggMo | 25 | 0.03 | $\beta = [0, 0.9, 0.99]$ |
| AggMo | 39 | 0.03 | |
| QHM | 25 | 1.0 | $\nu = 0.7, \beta = 0.999$ |
| QHM | 39 | 1.0 | |
| DEMON SGDM | 25 | 1.0 | $\beta_{\text{init}} = 0.5, \beta_{final} = -0.5$ |
| DEMON SGDM | 39 | 1.0 | $\beta_{\text{init}} = 0.3, \beta_{final} = -0.5$ |
| SGDM learning rate decay | 25 | 0.1 | $\beta_1 = 0.9$, smooth learning rate decay |
| SGDM learning rate decay | 39 | 1.0 | $\beta_1 = 0.0$, smooth learning rate decay |

Table 12: Best parameters for MNIST with VAE.

| Optimization method | epochs | $\eta$ | other parameters |
|---|---|---|---|
| Adam | 50 | 0.001 | |
| Adam | 100 | 0.001 | $\beta_1 = 0.9, \beta_2 = 0.999$ |
| Adam | 200 | 0.001 | |
| AMSGrad | 50 | 0.001 | |
| AMSGrad | 100 | 0.001 | $\beta_1 = 0.9, \beta_2 = 0.999$ |
| AMSGrad | 200 | 0.001 | |
| QHAdam | 50 | 0.001 | |
| QHAdam | 100 | 0.001 | $\nu_1 = 0.7, \nu_2 = 1.0, \beta_1 = 0.9, \beta_2 = 0.99$ |
| QHAdam | 200 | 0.001 | |
| DEMON Adam | 50 | 0.0001 | |
| DEMON Adam | 100 | 0.0001 | $\beta_{\text{init}} = 0.9, \beta_2 = 0.999$ |
| DEMON Adam | 200 | 0.0001 | |
| AggMo | 50 | 0.000003 | |
| AggMo | 100 | 0.000003 | $\beta = [0, 0.9, 0.99]$ |
| AggMo | 200 | 0.000003 | |
| QHM | 50 | 0.0001 | |
| QHM | 100 | 0.00003 | $\nu = 0.7, \beta = 0.999$ |
| QHM | 200 | 0.00003 | |
| DEMON SGDM | 50 | 0.00001 | |
| DEMON SGDM | 100 | 0.00001 | $\beta_{\text{init}} = 0.9$ |
| DEMON SGDM | 200 | 0.000003 | |
| SGDM learning rate decay | 50 | 0.00001 | $\beta_1 = 0.9$, patience = 5 |
| SGDM learning rate decay | 100 | 0.000003 | $\beta_1 = 0.9$, patience = 5 |
| SGDM learning rate decay | 200 | 0.000003 | $\beta_1 = 0.9$, patience = 20 |

## B  CONVERGENCE ANALYSIS

We analyze the global convergence of DEMON CM in the convex setting, following (Ghadimi et al., 2014). For an objective function $f$ which is convex, continuously differentiable, its gradient $\nabla f(\cdot)$ is Lipschitz continuous with constant $L$, our goal is to show that $f(\bar{\theta}_T)$ converges to the optimum $f^*$ with decreasing momentum, where $\bar{\theta}_T$ is the average of $\theta_t$ for $t = 1, ..., T$. Our following theorem holds for a constant learning rate and $\beta_t$ decaying with $t$.

**Theorem 1.** *Assume that $f$ is convex, continuously differentiable, its gradient $\nabla f(\cdot)$ is Lipschitz continuous with constant L, with a decreasing momentum, but constant step size, as in:*

$$\beta_t = \frac{1}{t} \cdot \frac{t+1}{t+2}, \quad \alpha \in \left(0, \frac{2}{3L}\right).$$

*We consider the CM iteration in non-stochastic settings, where:*

$$\theta_{t+1} = \theta_t - \alpha \nabla f(\theta_t) + \beta_t (\theta_t - \theta_{t-1}).$$

*Then, the sequence $\{\theta_t\}_{t=1}^T$ generated by the CM iteration, with decreasing momentum, satisfies:*

$$f(\bar{\theta}_T) - f^* \le \frac{\|\theta_1 - \theta^\star\|^2}{T} \left(\frac{3}{4} L + \frac{1}{2\alpha}\right),$$

*where $\bar{\theta}_T$ is the Cesaro average of the iterates: $\bar{\theta}_T = \frac{1}{T} \sum_{t=1}^T \theta_t$.*

*Proof.* Let $\beta_t = \frac{1}{t} \cdot \frac{t+1}{t+2}$ and

$$p_t = \frac{1}{t}(\theta_t - \theta_{t-1}).$$

We consider the CM iteration in non-stochastic settings, where:

$$\theta_{t+1} = \theta_t - \alpha \nabla f(\theta_t) + \beta_t (\theta_t - \theta_{t-1}).$$

Using the definition of $p_t$ above, one can easily prove that:

$$\theta_{t+1} + p_{t+1} = (1 + \frac{1}{t+1})\theta_{t+1} - \frac{1}{t+1}\theta_t = \theta_t + p_t - \frac{\alpha(t+2)}{t+1}\nabla f(\theta_t).$$

Using this expression, we will analyze the term $\|\theta_{t+1} + p_{t+1} - \theta^\star\|_2$:

$$\|\theta_{t+1} + p_{t+1} - \theta^\star\|^2 = \|\theta_t + p_t - \theta^\star\|^2 - \frac{2\alpha(t+2)}{t+1} \langle \theta_t + p_t - \theta^\star, \nabla f(\theta_t)\rangle + \left(\frac{\alpha(t+2)}{t+1}\right)^2 \cdot \|\nabla f(\theta_t)\|^2$$

$$= \|\theta_t + p_t - \theta^\star\|^2 - \frac{2\alpha(t+2)}{t(t+1)} \langle \theta_t - \theta_{t-1}, \nabla f(\theta_t)\rangle$$

$$- \frac{2\alpha(t+2)}{t+1} \langle \theta_t - \theta^\star, \nabla f(\theta_t)\rangle + \left(\frac{\alpha(t+2)}{t+1}\right)^2 \cdot \|\nabla f(\theta_t)\|^2$$

Since $f$ is convex, continuously differentiable, its gradient is Lipschitz continuous with constant L, then

$$\frac{1}{L}\|\nabla f(\theta_t)\|^2 \le \langle \theta_t - \theta^\star, \nabla f(\theta_t)\rangle, \tag{2}$$

$$f(\theta_t) - f^* + \frac{1}{2L}\|\nabla f(\theta_t)\|^2 \le \langle \theta_t - \theta^\star, \nabla f(\theta_t)\rangle, \tag{3}$$

$$f(\theta_t) - f(\theta_{t-1}) \le \langle \theta_t - \theta_{t-1}, \nabla f(\theta_t)\rangle. \tag{4}$$

Substituting the above inequalities leads to

$$\|\theta_{t+1} + p_{t+1} - \theta^\star\|^2 \le \|\theta_t + p_t - \theta^\star\|^2 - \frac{2\alpha(t+2)}{t(t+1)}(f(\theta_t) - f(\theta_{t-1}))$$

$$- 2\alpha \frac{(1-\lambda)(t+2)}{L(t+1)} \cdot \|\nabla f(\theta_t)\|^2 - 2\alpha\lambda\frac{t+2}{t+1}(f(\theta_t) - f^*)$$

$$- \left(\alpha\frac{\lambda(t+2)}{L(t+1)}\right) \cdot \|\nabla f(\theta_t)\|^2 + \left(\frac{\alpha(t+2)}{t+1}\right)^2 \cdot \|\nabla f(\theta_t)\|^2$$

where $\lambda \in (0, 1]$ is a parameter weighting (2) and (3). Grouping together terms yields

$$\left(\frac{2\alpha(t+2)}{t(t+1)} + \frac{2\alpha\lambda(t+2)}{t+1}\right)(f(\theta_t) - f^*) + \|\theta_{t+1} + p_{t+1} - \theta^\star\|^2$$

$$\le \frac{2\alpha(t+2)}{t(t+1)}(f(\theta_{t-1}) - f^*) + \|\theta_t + p_t - \theta^\star\|^2$$

$$+ \frac{\alpha(t+2)}{t+1}\left(\frac{\alpha(t+2)}{t+1} - \frac{2(1-\lambda)}{L} - \frac{\lambda}{L}\right)\|\nabla f(\theta_t)\|^2.$$

The last term is non-positive when $\alpha \in [0, \frac{t+1}{t+2}(\frac{2-\lambda}{L})]$ so it can be dropped. Summing over $t = 1, ..., T$ yields

$$2\alpha\lambda \sum_{t=1}^{T} \frac{t+2}{t+1}(f(\theta_t) - f^*) + \sum_{t=1}^{T} \left( \frac{2\alpha(t+2)}{t(t+1)}(f(\theta_t) - f^*) + \|\theta_{t+1} + p_{t+1} - \theta^\star\|^2 \right)$$
$$\leq \sum_{t=1}^{T} \left( \frac{2\alpha(t+2)}{t(t+1)}(f(\theta_{t-1}) - f^*) + \|\theta_t + p_t - \theta^\star\|^2 \right),$$

implying that:

$$2\alpha\lambda \sum_{t=1}^{T} \frac{t+2}{t+1}(f(\theta_t) - f^*) \leq 3\alpha(f(\theta_1) - f^*) + \|\theta_1 - \theta^\star\|^2.$$

Since:

$$2\alpha\lambda \sum_{t=1}^{T} (f(\theta_t) - f^*) \leq 2\alpha\lambda \sum_{t=1}^{T} \frac{t+2}{t+1}(f(\theta_t) - f^*) \leq 3\alpha\lambda \sum_{t=1}^{T} (f(\theta_t) - f^*),$$

we further have:

$$3\alpha\lambda \sum_{t=1}^{T} (f(\theta_t) - f^*) \leq \tfrac{3}{2}\left( 3\alpha(f(\theta_1) - f^*) + \|\theta_1 - \theta^\star\|^2 \right).$$

Due to the convexity of $f$,

$$f(\bar{\theta}_t) \leq \tfrac{1}{T} \sum_{t=1}^{T} f(\theta_t),$$

observe that

$$f(\bar{\theta}_T) - f^* \leq \tfrac{1}{T} \sum_{t=1}^{T} (f(\theta_t) - f^*) \leq \tfrac{1}{3\alpha\lambda T} \left( \tfrac{9}{2}\alpha(f(\theta_1) - f^*) + \tfrac{3}{2}\|\theta_1 - \theta^\star\|^2 \right).$$

Since $f(\theta_1) - f^* \leq \frac{L}{2}\|\theta_1 - \theta^\star\|^2$ by Lipschitz continuous gradients, setting $\lambda = 1$ and observing $(t + 1)/(t + 2) \geq 2/3$ gives the result.

## C  ADDITIONAL PLOTS

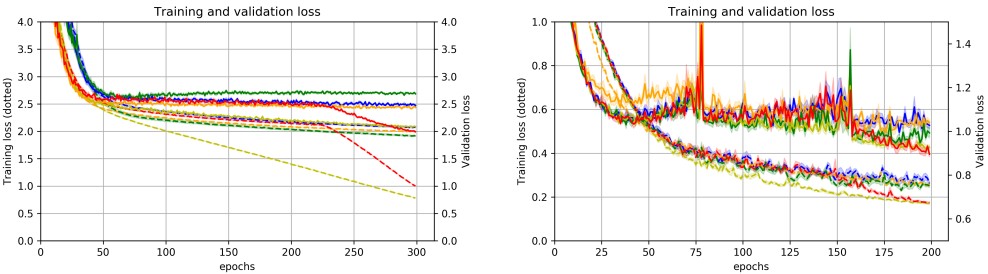

Figure 4: Additional empirical results on adaptive learning rate methods. Left plot: `VGG16-CIFAR100-DEMONAdam` for 300 epochs. Right plot: `WRN-STL10-DEMONAdam` for 200 epochs. Dotted and solid lines represent training and generalization metrics respectively. Shaded bands represent 1 standard deviation.

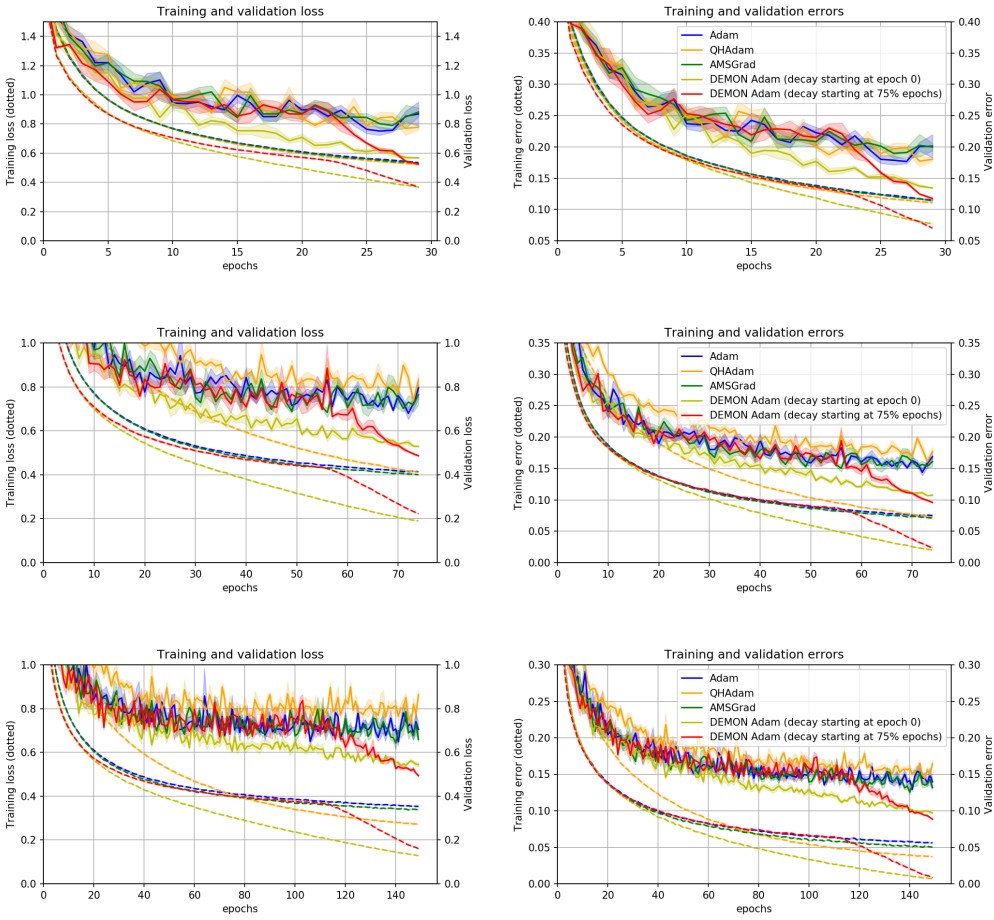

Figure 5: Additional empirical results on `RN18-CIFAR10-DEMONAdam`. Top row: 30 epochs. Middle row: 75 epochs. Bottom row: 150 epochs. Dotted and solid lines represent training and generalization metrics respectively. Shaded bands represent 1 standard deviation.

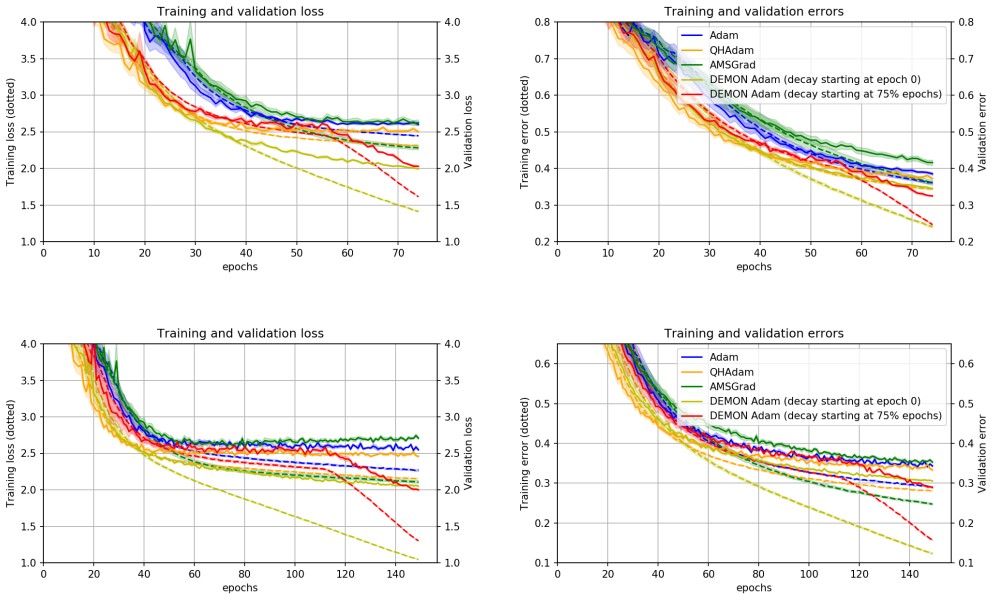

Figure 6: Additional empirical results on `VGG16-CIFAR100-DEMONAdam`. Top row: 75 epochs. Bottom row: 150 epochs. Dotted and solid lines represent training and generalization metrics respectively. Shaded bands represent 1 standard deviation.

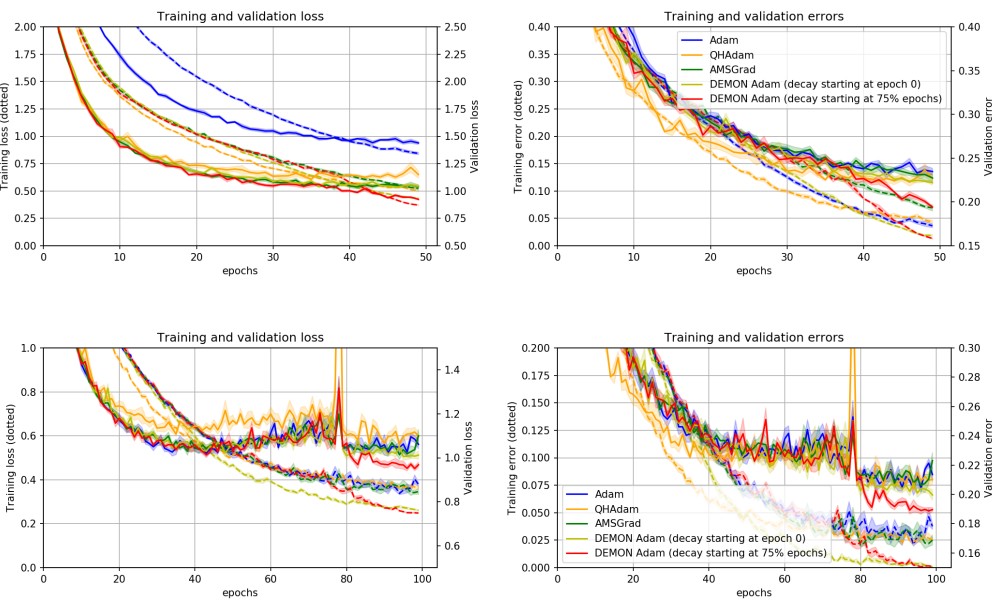

Figure 7: Additional empirical results on `WRN-STL10-DEMONAdam`. Top row: 50 epochs. Bottom row: 100 epochs. Dotted and solid lines represent training and generalization metrics respectively. Shaded bands represent 1 standard deviation.

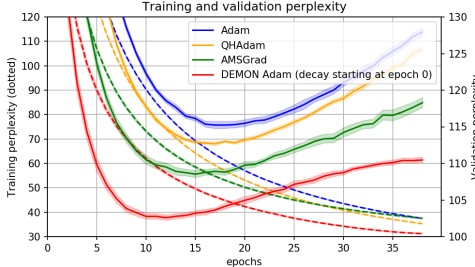

Figure 8: Additional empirical results on `PTB-LSTM-DEMONAdam` for 39 epochs. Dotted and solid lines represent training and generalization metrics respectively. Shaded bands represent 1 standard deviation.

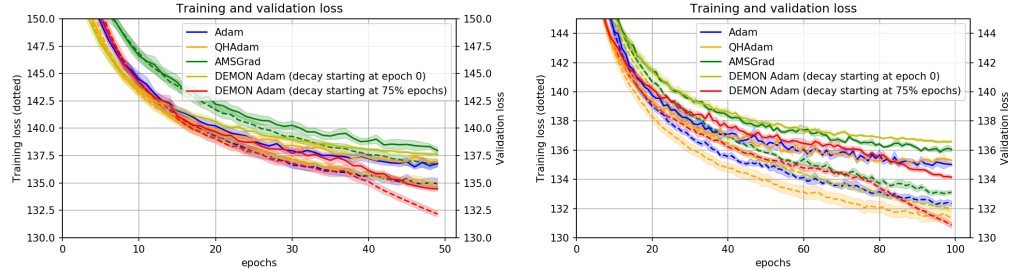

Figure 9: Additional empirical results on `VAE-MNIST-DEMONAdam`. Left: 50 epochs. Right: 100 epochs. Dotted and solid lines represent training and generalization metrics respectively. Shaded bands represent 1 standard deviation.

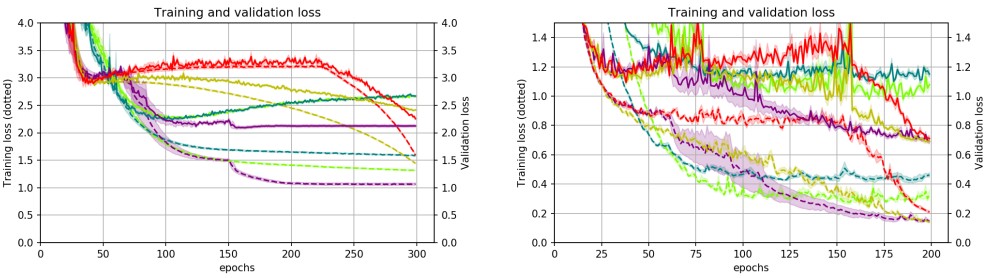

Figure 10: Additional empirical results on adaptive momentum methods. Left plot: `VGG16-CIFAR100-DEMONSGDM` for 300 epochs. Right plot: `WRN-STL10-DEMONSGDM` for 200 epochs. Dotted and solid lines represent training and generalization metrics respectively. Shaded bands represent 1 standard deviation.

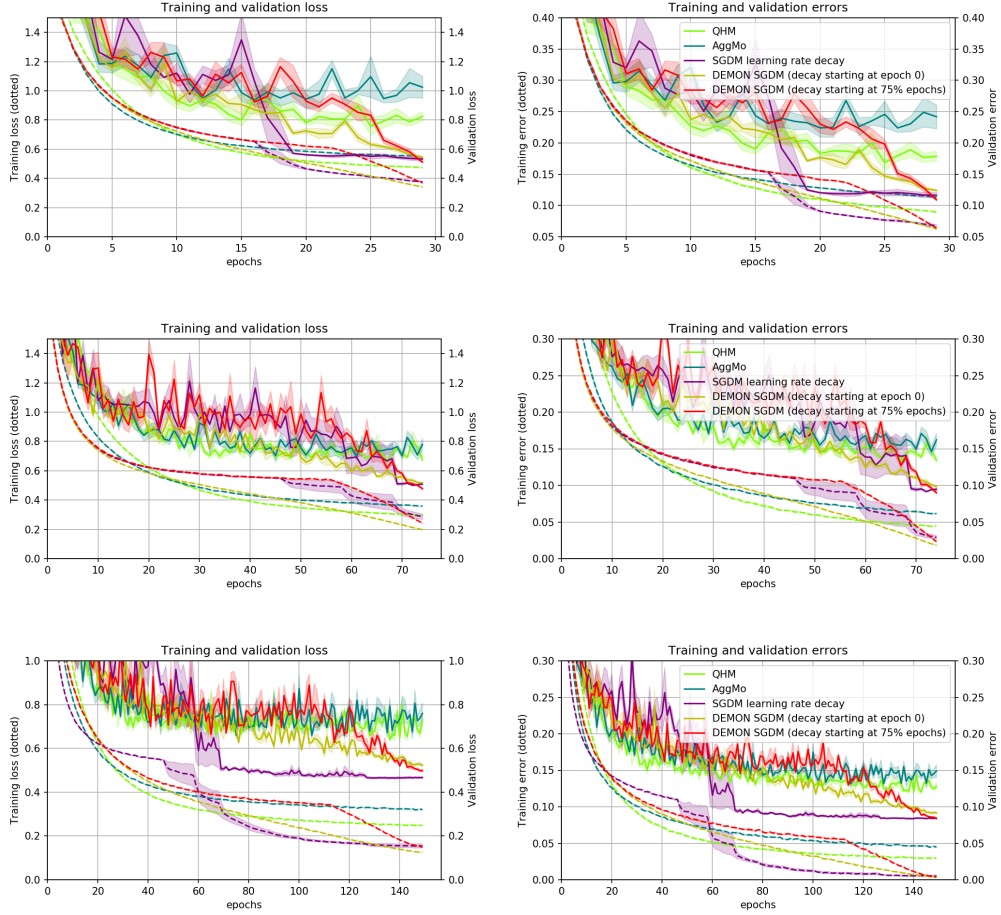

Figure 11: Additional empirical results on `RN18-CIFAR10-DEMONSGDM`. Top row: 30 epochs. Middle row: 75 epochs. Bottom row: 150 epochs. Dotted and solid lines represent training and generalization metrics respectively. Shaded bands represent 1 standard deviation.

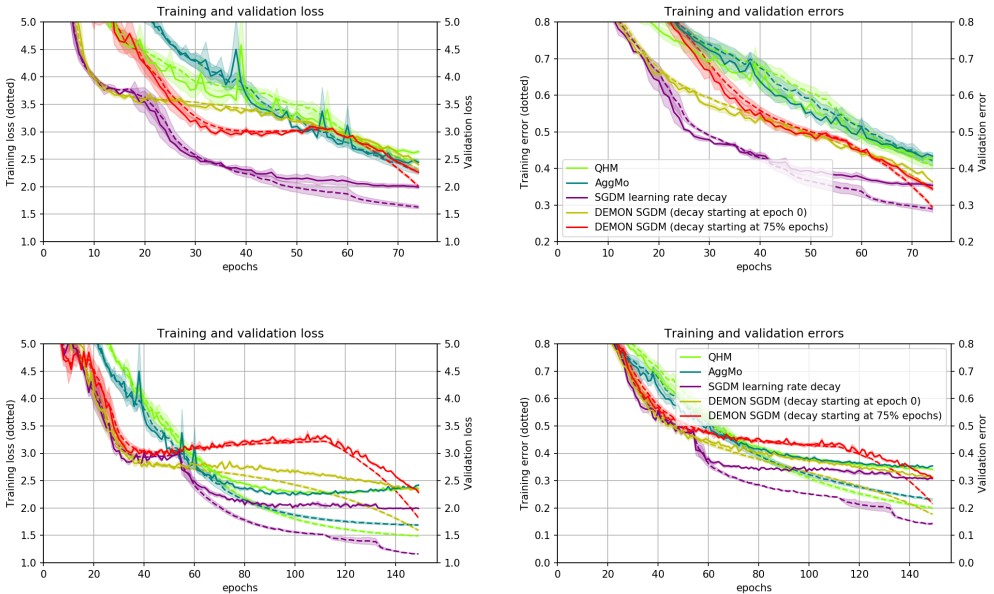

Figure 12: Additional empirical results on `VGG16-CIFAR100-DEMONSGDM`. Top row: 75 epochs. Bottom row: 150 epochs. Dotted and solid lines represent training and generalization metrics respectively. Shaded bands represent 1 standard deviation.

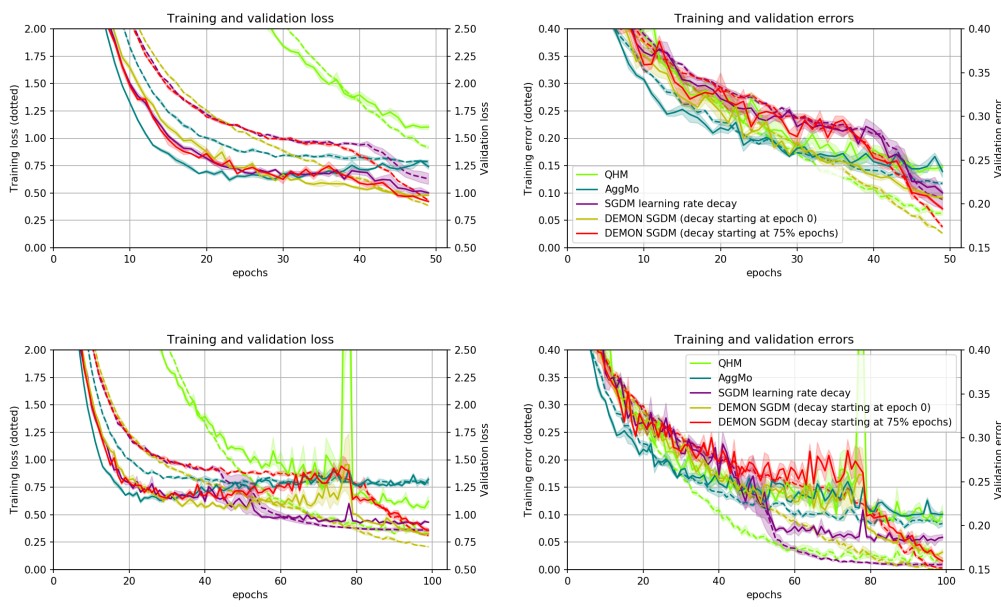

Figure 13: Additional empirical results on `WRN-STL10-DEMONSGDM`. Top row: 50 epochs. Bottom row: 100 epochs. Dotted and solid lines represent training and generalization metrics respectively. Shaded bands represent 1 standard deviation.

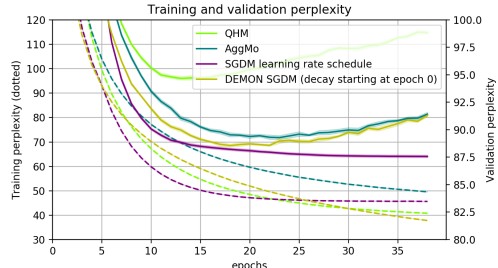

Figure 14: Additional empirical results on `PTB-LSTM-DEMONSGDM` for 39 epochs. Dotted and solid lines represent training and generalization metrics respectively. Shaded bands represent 1 standard deviation.

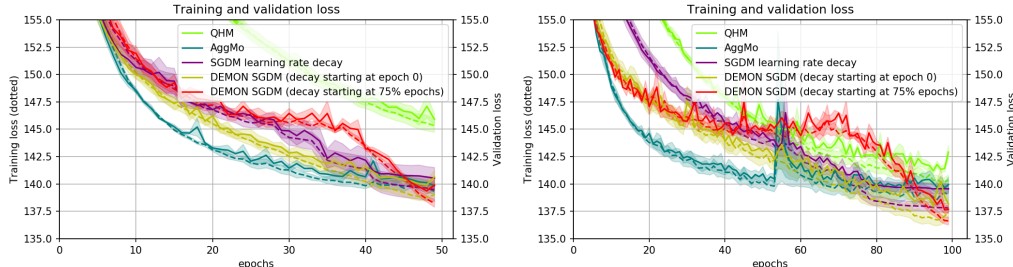

Figure 15: Additional empirical results on `VAE-MNIST-DEMONSGDM`. Left: 50 epochs. Right: 100 epochs. Dotted and solid lines represent training and generalization metrics respectively. Shaded bands represent 1 standard deviation.

