# OpenReview forum: "Decaying momentum helps neural network training"
_ICLR.cc/2020/Conference — Reject_

### Official Review · AnonReviewer2 · 2019-10-22
**Official Blind Review #2**

**Rating:** 3

**Review:**

The authors propose Demon schedule that automatically decays momentum. The paper is very well written, and the experiments are performed on an, extensive, compared to typical papers in the domain, suite of tasks. Unfortunately, while the paper is technically well executed, I have fundamental issues with novelty. Based on the current state-of-the-art understanding of optimization in deep learning, it is quite expected that decaying momentum has an analogous effect to decaying learning rate or increasing batch size (see [1,2]). Given this "Similarly, applying DEMON to momentum SGD rivals momentum SGD with learning rate decay" is not surprising, and "and in many cases leads to improved performance" warrarnts a bit of scepticism.

More precisely, a very similar analysis as in "Motivation and interpretation." can be already found in [1]. Similarly, [2] already suggests decreasing momentum. Based on this, experiments in Table. 2 (and the other similar Tables) should compare to decaying learning rate or batch size in Adam and other adaptive methods using an analogous schedule. It might be surprisng, but adaptive methods do benefit from learning rate schedules. Analogously, could you please compare performance of Demon to the recommendation in [2] to decay momentum together with increasing learning rate initially?

Unless a more detailed experiments demonstrate that tuning momentum brings additional benefit on top of its effect on the effective learning rate, there isn't unfortunately in my onion enough practical value in the proposed method.

References:

[1] Smith et al, Don't Decay the Learning Rate, Increase the Batch Size, https://arxiv.org/abs/1711.00489
[2] Fast.ai documentation on one cycle method, https://docs.fast.ai/callbacks.one_cycle.html

**Experience Assessment:**

I have published one or two papers in this area.

**Review Assessment: Checking Correctness Of Derivations And Theory:**

I assessed the sensibility of the derivations and theory.

**Review Assessment: Checking Correctness Of Experiments:**

I assessed the sensibility of the experiments.

**Review Assessment: Thoroughness In Paper Reading:**

I read the paper at least twice and used my best judgement in assessing the paper.

---

> ### Author Response · Authors · 2019-11-08
> **Response**
>
> Thank you for your review.
>
> ''...it is quite expected that decaying momentum has an analogous effect to decaying learning rate or increasing batch size ... a very similar analysis as in "Motivation and interpretation." can be already found in [1]. Similarly, [2] already suggests decreasing momentum.''
>
> >> Thank you for the link to the works and we have added them as relevant citations and introduced them in section 4, the related works section. Regarding novelty, [1] may have similar analysis, but their proposal is totally different, suggesting increasing momentum. [2] requires more hyperparameters, including defining maximum and minimum learning rates and momentum. Also, it suggests one or more cycles of simultaneously increasing learning rate and decreasing momentum, and then simultaneously decreasing learning rate and increasing momentum, which is quite different from our proposal. We want to emphasize that Demon is a practical alternative with no additional hyperparameter tuning. In particular, the way the momentum decays is critical to producing strong performance, and has not been extensively explored before.
>
> ''...adaptive methods do benefit from learning rate schedules.''
>
> >> We point out that for the case of learning rate decay, the smooth learning rate decay implemented in major libraries produces bad performance, as opposed to Demon. Nevertheless, following your suggestion, we have reworded some of our introduction to be more conservative.
> Decaying the learning rate for adaptive learning rate algorithms is additional tuning and for many settings this defeats the purpose of adaptive algorithms, since hand-tuned momentum SGD is often superior. Demon requires no additional tuning so this is a fair comparison.
>
> ''...compare performance of Demon to the recommendation in [2] to decay momentum together with increasing learning rate initially?''
>
> >> We conducted preliminary studies of 1cycle with momentum SGD for the settings of ResNet18 on CIFAR10 for 300 epochs and VGG16 on CIFAR100 for 150 epochs.
>
> Following the suggestions in the paper, for RN18-CIFAR10 we try all combinations of maximum learning rate = [3.0,1.0,0.3,0.1,0.03,0.01], maximum momentum = [0.97,0.95,0.9], minimum momentum=[0.85,0.8], batch size=[128,256,512], with minimum learning rate=0.1 * maximum learning rate. The lowest test error is achieved with maximum learning rate=1.0, maximum momentum=0.95, minimum momentum=0.85, batch size = 512, achieving 7.65 +- .13. Demon CM, with no tuning, achieves 7.58 +- .04.
>
> For VGG16-CIFAR100, we try all combinations of maximum learning rate = [1.0,0.3,0.1,0.03], maximum momentum = [0.97,0.95,0.9], minimum momentum=[0.85,0.8], batch size=[128,256,512], with minimum learning rate=0.1 * maximum learning rate. The lowest test error is achieved with maximum learning rate=0.1, maximum momentum=0.95, minimum momentum=0.85, batch size = 512, achieving 32.05 +- 1.05. In comparison, Demon CM, with no tuning, achieves significantly lower at 30.59 +- .26.
>
> As you can see, 1cycle with momentum SGD shows strong performance but there is still a small performance gap, in addition to AggMo, QHM and momentum SGD in the paper, with respect to Demon CM. If accepted, following your advice we will add experiments for 1cycle for all the settings considered in the paper for completeness for the camera-ready version. Thanks again for the link to the works and we have cited both works and mentioned them in the related works section.
>
> ''Unless a more detailed experiments demonstrate that tuning momentum brings additional benefit on top of its effect on the effective learning rate, there isn't unfortunately in my onion enough practical value in the proposed method.''
>
> >> The primary goal of this paper is to present an easy-to-tune method with demonstrated improved performance. The motivation, interpretation, and method is quite different from [1, 2]. We have demonstrated the effectiveness of Demon across a wide variety of tasks and against many modern optimizers and have shown preliminary results with 1cycle with momentum SGD as well. Demon also works for both adaptive learning rate methods and non-adaptive learning rate methods. Lastly and most importantly, Demon requires no additional tuning and limited extra overhead. This is an effective and practical method which is the goal of our paper.

---

> > ### Comment · AnonReviewer2 · 2019-11-15
> > **Thank you for your rebuttal**
> >
> > Thank you for your rebuttal. I would like to uphold my score on the following grounds:
> >
> > (1) It seems to me that Demon offers a similar perfomance to one-cycle. There is a "small" performance gap, but it is indeed very small.
> >
> > (2) One-cycle is not that much harder to tune. It does have maximum and minimum learning rate, but these can be quite quickly adapted using "learning rate finder" heuristic (see fast.ai blog).
> >
> > (3) There is an issue with lack of novelty compared to [1]. I am confused by  "[1] may have similar analysis, but their proposal is totally different, suggesting increasing momentum. ". [1]  main contribution is that increasing momentum is equivalent to increasing learning rate, or decreasing batch size.  Hence, a correct baseline would be comparing Demon to the equivalent learning rate schedule.
> >
> > At the same time, I really liked the extensive experiment setting, and wouldn't be upset to see the paper accepted if other reviewers champion it. My opinion is that a rewrite that better compares to state of the art is needed. Another direction would be a closer analysis for why momentum and learning rate are equivalent. Arguably, both Demon's, and [1] analysis is very simplistic.

---

> > > ### Author Response · Authors · 2019-11-15
> > > **Response**
> > >
> > > Thank you for your response.
> > >
> > > (1) It seems ... it is indeed very small.
> > >
> > > >> While for our experiments for the ResNet18 CIFAR10 setting the gap is small, the gap for VGG16 CIFAR100 is substantial (1.5%) gap, given the very optimized performance of existing algorithms.
> > >
> > > (2) One-cycle is ... using "learning rate finder" heuristic (see fast.ai blog).
> > >
> > > >> We would like to add that we have a convergence proof for decaying momentum for convex problems, while One-cycle has no proof of convergence.
> > >
> > > (3) There is an issue ... the equivalent learning rate schedule.
> > >
> > > >> In our understanding, [1] primarily makes the following suggestions: 1. An equivalence between learning rate decay and batch size increase. 2. Increasing learning rate and scaling batch size, or increasing momentum and batch size. Both are different from our proposal, and our proposal requires no batch size or learning rate scheduling. In addition, the large batch case in the paper is not always practical due to computational storage limits.

---

### Official Review · AnonReviewer1 · 2019-10-23
**Official Blind Review #1**

**Rating:** 3

**Review:**

This paper proposed a new decaying momentum rule to further improve neural network training. The idea is motivated by decaying the total contribution of a gradient to all future updates. The authors also extend this idea on to Adam and show that it improved upon the vanilla Adam.

- The intuition of using momentum decaying scheme is not quite clear to me. Why having a linear momentum decay schedule is better? This question is not answered in the paper. I hope the authors could provide more illustrative explanations regarding this.

- This paper focused essentially on empirical evaluations. I do appreciate that the authors conduction various experiments using different dataset and architecture but I do not understand why the authors want to separate the experiments into parts: adaptive methods and adaptive momentum methods. Not to mention the confusing names, it makes no sense to say the same words again for different optimizers. I would suggest the authors to combine the results together for better comparison.

- CM formulation mentioned in the paper is actually different from the commonly refer SGD with momentum implementation, which is the optimizer that is widely used in the community. Therefore, it is important to at least add SGD with momentum as one baseline in the experiments.

- According to https://github.com/kuangliu/pytorch-cifar, Resnet18 on CIFAR10 can achieve at least 93% accuracy using simple SGD with momentum. It seems that the current reported results are not fully optimized. I would suggest the authors to check the parameter settings and make sure all the hyperparameters for baseline methods are fully tuned.

- Aside from SGD with momentum, the experiments part also lacks several important baselines. The author should also consider comparing the AdamW, Padam mentioned in the paper.


**Experience Assessment:**

I have published in this field for several years.

**Review Assessment: Checking Correctness Of Derivations And Theory:**

I carefully checked the derivations and theory.

**Review Assessment: Checking Correctness Of Experiments:**

I carefully checked the experiments.

**Review Assessment: Thoroughness In Paper Reading:**

I read the paper thoroughly.

---

> ### Author Response · Authors · 2019-11-08
> **Response part 1**
>
> ``Thank you for your review.
>
> ''The intuition of using momentum ... explanations regarding this.''
>
> >> This decay scheme requires no tuning--since we primarily decay to 0--works well empirically, and is intuitive by Occam's Razor. We also borrow ideas from learning rate linear decaying models. We do note that the way momentum is decayed is critical to good performance, and the smooth decaying learning rate schemes implemented in major libraries also perform significantly worse. Following your advice, we have emphasized this in the motivation and interpretation section of the paper and improved clarity.
>
> ''This paper focused essentially...combine the results together for better comparison.''
>
> >> With regards to splitting the experiments, we follow [1] where they separately compare adaptive learning rate algorithms and non-adaptive learning rate algorithms, even for experiments on the same task. In many settings, adaptive learning rate algorithms produce worse performance than hand-tuned non-adaptive learning rate algorithms, but are used because they reduce the need for hyperparameter tuning. Thus, it makes sense to compare them separately. Also, for practicality it is difficult to parse the plots due to the many curves, if we include all methods in one plot, since there are 8 methods already and including the suggested Padam and AdamW experiments would lead to 10 methods. With regards to same words for different optimizers, we have corrected the use of CM, momentum SGD, and accelerated SGD to be consistent across the paper to improve clarity. We are not aware of any other cases of this and thank you for catching that.
>
> ''``CM formulation...baseline in the experiments.''
>
> >> ***For our experiments with momentum SGD, we actually use the official implementation in PyTorch and TensorFlow which is the implementation you are referring to!*** The particular CM formulation in the introduction refers to the fixed learning rate case, and in that case it is equivalent. However, this is a good point and we have changed the formula in our paper to correct this difference.
>
> ''``According to https://github.com/kuangliu/pytorch-cifar, Resnet18 on CIFAR10 can achieve at least 93\% accuracy.''
>
> >> We used standard hyperparameter tuning methods in practice for tuning SGD with momentum. The commonly used practical variant of learning rate decay is to decay on plateau of validation error and this is the one we include in our paper, and even so this requires some tuning which we do. Demon is a practical alternative which requires no tuning, and fine tuning Demon leads to further improvements in performance. Following your suggestion, we ran the code at the link for 350 epochs, the number of epochs prescribed at the link, and with slight fine-tuning with Demon CM were able to achieve 95\% test accuracy.
>
> ''``Aside from SGD with momentum, the experiments part also lacks several important baselines. The author should also consider comparing the AdamW, Padam mentioned in the paper.''
>
> >> We note that we did not intentionally leave out these algorithms from our comparison list: according to QHAdam paper [1], QHAdam is capable recovering both AdamW and Padam. Thus we excluded those from the comparison list. We also have edited the text to reflect this comment.
>
> However, following your advice, we conducted preliminary studies of AdamW and Padam for the settings of ResNet18 on CIFAR10 with 300 epochs, VGG16 on CIFAR100 with 150 epochs, and Variational AutoEncoder on MNIST with 50 epochs.
>
> For RN18-CIFAR10, following the recommendations in the papers we tried all combinations of the following parameters for AdamW, learning rate in [0.01,0.003,0.001,0.0003], weight decay in [0.01,0.003, 0.001,0.0003,0.0001,0.00003], $\beta_1=0.9$, $\beta_2=0.999$. The lowest test error is attained with learning rate=0.001 and weight decay=0.0001, at 11.34 +- .83. Following the Padam paper, we try learning rate in [0.1,0.03,0.01,0.003,0.001,0.0003], p in [1/4,1/8,1/16], $\beta_1=0.9$, $\beta_2=0.999$. The lowest test error is attained with learning rate=0.01 and p=1/4, at 12.13 +- .70. Demon Adam achieves significantly lower test error at 8.44 +- .05.
>
> For VGG16-CIFAR100, we tried all combinations of the following parameters for AdamW, learning rate in [0.003,0.001,0.0003,0.0001], weight decay in [0.01,0.003, 0.001,0.0003,0.0001,0.00003], $\beta_1=0.9$, $\beta_2=0.999$. The lowest test error is achieves with learning rate=0.0003 and weight decay=0.001, at 33.71 +- .23. For Padam, we try learning rate in [0.1,0.03,0.01,0.003,0.001,0.0003], p in [1/4,1/8,1/16], $\beta_1=0.9$, $\beta_2=0.999$. The lowest test error is attained with learning rate=0.03 and p=1/16, at 34.38 +- .71. Demon Adam, again, achieves significantly lower test error at 28.84 +- .18.
>
> [1] Jerry Ma and Denis Yarats. Quasi-hyperbolic momentum and Adam for deep learning. arXiv preprint arXiv:1810.06801, 2018.

---

> > ### Comment · AnonReviewer1 · 2019-11-14
> > **Official Blind Review #1**
> >
> > Thank you for your response.
> >
> > I notice that you change the formulation for CM in the revision. This formulation is now align with SGD with momentum in Pytorch/Tensorflow, but is not CM in (Polyak, 1964). CM and SGD with momentum are two different algorithms and I am confused which algorithm the authors are trying to compare?
> >
> > I appreciate the your effort to made the extra comparisons. However, the reported number seems to be much lower than the original papers under the same dataset/architecture. I wonder if the authors only apply learning rate decay to “CM” (or SGD?) ? The same learning rate decay schedule should be applied to all methods for a fair comparison.

---

> > > ### Author Response · Authors · 2019-11-14
> > > **Response**
> > >
> > > Thank you for your review.
> > >
> > > "I notice that you change the ... are trying to compare?"
> > >
> > > >> We employ the momentum SGD you are referring to and that is now mentioned in the text. The two formulations are equivalent in the fixed learning rate scenario. However, following your advice and to prevent any further confusion, we've made some wording changes in the manuscript to be consistent to SGD with momentum.
> > >
> > > "I appreciate the your effort to ... applied to all methods for a fair comparison."
> > >
> > > >> We don't decay the learning rate for the adaptive learning rate algorithms in the previous response because this is additional tuning and for many settings defeats the purpose of using adaptive algorithms, since in that case oftentimes hand-tuned momentum SGD is superior. Meanwhile, Demon requires no additional tuning. Furthermore, for Padam we have completed additional hyperparameter tuning already for the additional p parameter. With regards to performance, the ResNet model used in AdamW is deeper and different from the one used in our paper. For Padam, they employ two extra hyperparameter tuning steps of using a learning rate schedule and tuning p, and other subtle differences may exist in data augmentation and architecture. We stress again that Demon requires no additional tuning, and in many settings any additional learning rate decay tuning defeats the purpose of using adaptive methods since hand-tuned momentum SGD is often better.

---

> ### Author Response · Authors · 2019-11-08
> **Response part 2**
>
> For VAE-MNIST, we tried all combinations of the following parameters for AdamW, learning rate in [0.003,0.001,0.0003,0.0001], weight decay in [0.01, 0.003, 0.001, 0.0003, 0.0001, 0.00003, 0.00001], and $\beta_1 = 0.9$, $\beta_2=0.999$. The lowest validation loss is attained with learning rate = 0.001 and weight decay = 0.0001, with a loss value of 136.51 +- .39. For Padam, learning rate in [0.003,0.001,0.0003,0.0001], exponent p in [0.4, 0.25, 0.125, 0.0625] and $\beta_1 = 0.9$, $\beta_2=0.999$. The lowest validation loss is attained with learning rate = 0.001 and p=0.4, with a loss value of 137.37 +- .75. For this task, Demon Adam achieves 134.46 +- .17, substantially better than AdamW and Padam.
>
> As you can see, there remains a performance gap between AdamW and Padam, in addition to AMSGrad, QHAdam and Adam in the paper, with respect to Demon Adam. We also don't rule out improving AdamW and Padam with Demon, but we have only applied Demon to Adam and momentum SGD in this paper because they are the most widely used. If accepted, following your advice we will add experiments for AdamW and Padam for all the settings considered in the paper for completeness for the camera-ready version.

---

### Official Review · AnonReviewer3 · 2019-10-23
**Official Blind Review #3**

**Rating:** 6

**Review:**

In this paper, the author propose a decaying momentum rule to improve algorithm. Furthermore, he apply this rule in momentum SGD and Adam, then use experiment to prove the algorithm.

In the experiment, it training on many different dataset and compare with many baseline. The algorithm with decaying momentum rule get much better result than all other algorithm. Furthermore, the paper is well written and east to follow.

However, I still have a question about this paper.

Even the algorithm get a good performance, the intuition of the algorithm is not clear. In fact, it is reasonable to decay the total contribution of a gradient to all future updates; it is still unclear why you choose such $\beta$. As you say, the contribution of previous item is $ \sum \beta^i=\frac \beta {(1-\beta)} $, however, this is only correct why $\beta$ is constant, and when the $\beta_t$ is change like your definition, what will the contribution become? Furthermore, even the contribution of previous item is $ \frac \beta {(1-\beta)} $, why you want the contribution is equal to $ (1-\frac t T)\beta_{init}/(1-\beta_{init})$, it may need more comment about why you choose the value.


**Experience Assessment:**

I have read many papers in this area.

**Review Assessment: Checking Correctness Of Derivations And Theory:**

I assessed the sensibility of the derivations and theory.

**Review Assessment: Checking Correctness Of Experiments:**

N/A

**Review Assessment: Thoroughness In Paper Reading:**

I read the paper at least twice and used my best judgement in assessing the paper.

---

> ### Author Response · Authors · 2019-11-08
> **Response**
>
> Thank you for your review.
>
> >> We understand your point. To address your concern, $\beta_t$ tends to change slowly with respect to a particular $g_i$ and thus it is a very close approximation. Consider for $\beta_i = 0.9$ and the setting of 300 epochs on CIFAR10 with a batch size of 128, yielding roughly 100,000 iterations. At any $i$, after 50 iterations $\beta$ barely changes but the gradient has almost "disappeared" since $(\beta_{i} \beta_{i + 1} \dots \beta_{t}) \approx 0.9^{50} \approx 0.005$. By the time $\beta$ has changed substantially $g_i$ is already negligible so $\beta_i / (1 - \beta_i)$ is a very close approximation. With regards to $(1 - \frac{t}{T}) \beta_{init} / (1 - \beta_{init})$, we wish to decay this value linearly so we pick $\beta_t$ such that $\beta_t / (1 - \beta_t) = (1 - \frac{t}{T}) \beta_{init} / (1 - \beta_{init})$, where $(1 - \frac{t}{T})$ is the proportion of total iterations remaining. We will make this clearer in our paper. This is a simple scheme by Occam's Razor which requires no tuning by decaying to 0 and shows strong performance. Thank you for the pointers, following your advice we have clarified those points in section 3 in our paper for improved clarity and understanding.

---

> > ### Comment · AnonReviewer3 · 2019-11-14
> > **Official Blind Review #3**
> >
> > I see the revision and I think the idea is more reasonable. However, I am not familiar with the experiment and I am not sure whether the comment of review 3 is correct. If the experiment part is correct, I will increase my score.

---

> > > ### Author Response · Authors · 2019-11-14
> > > **Response**
> > >
> > > >> Thank you for your review. Let us expand a little on the previous comment. For the particular example of CIFAR-10, 300 epochs and a batch size of 128 is a fairly common setup, such as in the ResNeXt paper [1]. There are 60k total images in that dataset, and typically 50k are used for training, leading to $50,000 * 300 / 128 = 117187$ total iterations. Referring to the previous response, in the context of the total number of iterations $\beta_i / (1 - \beta_i)$ is a very close approximation.
> > >
> > > [1] Saining Xie, Ross B. Girshick, Piotr Dollar, Zhuowen Tu, and Kaiming He. Aggregated residual transformations for deep neural networks. In CoRR, abs/1611.05431, 2016.

---

### Decision · Program_Chairs · 2019-12-19

**Decision:**

Reject

**Comment:**

This paper proposes a new decaying momentum rule to improve existing optimization algorithms for training deep neural networks, including momentum SGD and Adam. The main objections from the reviewers include: (1) its novelty is limited compared with prior work; (2) the experimental comparison needs to be improved (e.g., the baselines might not be carefully tuned, and learning rate decay is not applied, while it usually boosts the performance of all the algorithms a lot). After reviewer discussion, I agree with the reviewers’ evaluation and recommend reject.